# Large interannual variability in supraglacial lakes around East Antarctica

Jennifer F. Arthur [1✉], Chris R. Stokes [1], Stewart S. R. Jamieson [1], J. Rachel Carr[2], Amber A. Leeson [3] & Vincent Verjans [4]

Antarctic supraglacial lakes (SGLs) have been linked to ice shelf collapse and the subsequent acceleration of inland ice flow, but observations of SGLs remain relatively scarce and their interannual variability is largely unknown. This makes it difficult to assess whether some ice shelves are close to thresholds of stability under climate warming. Here, we present the first observations of SGLs across the entire East Antarctic Ice Sheet over multiple melt seasons (2014–2020). Interannual variability in SGL volume is >200% on some ice shelves, but patterns are highly asynchronous. More extensive, deeper SGLs correlate with higher summer (December-January-February) air temperatures, but comparisons with modelled melt and runoff are complex. However, we find that modelled January melt and the ratio of November firn air content to summer melt are important predictors of SGL volume on some potentially vulnerable ice shelves, suggesting large increases in SGLs should be expected under future atmospheric warming.

[1] Department of Geography, Durham University, Durham DH1 3LE, UK. [2] School of Geography, Politics and Sociology, Newcastle University, Newcastle-upon-Tyne NE1 7RU, UK. [3] Lancaster Environment Centre/Data Science Institute, Lancaster University, Bailrigg, Lancaster LA1 4YW, UK. [4] School of Earth and Atmospheric Sciences, Georgia Institute of Technology, Atlanta, GA, USA. ✉email: jennifer.arthur@durham.ac.uk

Supraglacial lakes (SGLs) have been linked to Antarctic ice-shelf disintegration[1–6] and the subsequent acceleration of grounded ice inland, increasing mass loss and contributing to sea-level rise[7–10]. Of concern is that SGLs are expected to become more extensive on Antarctic ice shelves, due to increases in surface melt extent and intensity in response to future atmospheric warming[11–14]. Regular, prolonged surface melt reduces the meltwater retention capacity of ice shelves by saturating their firn layer and reducing firn air content (FAC)[11,15–18]. Excess meltwater that cannot be stored in the firn runs off to form SGLs on the snow or ice surface, filling topographic hollows, including rifts and crevasses[16]. Observations and modelling have linked such SGLs on ice shelves to meltwater-induced vertical fracture propagation, termed hydrofracturing[3,4,6,19–21]. Some regions of ice shelves are already vulnerable to hydrofracturing[18,20], but ice shelves around Antarctica will become more prone to hydrofracture-driven break-up as surface melt continues to lower their ability to accommodate meltwater[11,12,14,18,22,23]. Furthermore, and despite uncertainties, numerical models that attempt to capture these processes show much higher sea-level contributions from Antarctica, due to earlier ice shelf removal and loss of buttressing[13,24]. Thus, there is an urgent need to better constrain where and when SGLs develop on Antarctic ice shelves and which ice shelves are closer to potential thresholds in meltwater-induced hydrofracturing than others[14,18,20].

Recent assessments using satellite observations have found that SGLs are more extensive than previously thought on the world's largest ice sheet in East Antarctica, which holds the vast majority of the Earth's glacier ice (~52 m of sea level equivalent)[15,23]. Studies on a handful of individual ice shelves have also quantified the seasonal evolution of SGLs in East Antarctica[22,25–29]. However, there is limited knowledge of the interannual variability in SGLs over multiple melt seasons and across the whole ice sheet. There have also been few attempts to link the spatial and temporal variability in SGLs on ice shelves to near-surface climatic conditions[22,26,30,31]. Here, we quantify the variability in SGL distributions and volumes around the East Antarctic Ice Sheet (EAIS) during the peak of seven consecutive melt seasons (2014–2020) and investigate potential climatic controls on their development and near-surface (i.e. firn) conditions generated by ERA5 climate reanalysis and the Community Firn Model (CFM)[32] forced by the regional climate model MARv3.11[33].

## Results

We apply a threshold-based algorithm[34] to Landsat 8 satellite imagery around the EAIS margin during January (which is known to be the peak of the melt season)[2,26,28,29] from 2014 to 2020 (Fig. 1, Methods). Our focus is on supraglacial lakes (SGLs) due to their key role in hydrofracturing[2,3,6,20] but we recognise that these often co-exist with surface streams and areas of slush[15,23,27,28].

The peak in total (EAIS-wide) SGL volume occurred in January 2017 (2331 ×10^6 m^3) (Fig. 2a), linked to large positive anomalies on the Roi Baudouin (Fig. 2e) and Amery (Fig. 2f) ice shelves, which together accounted for 80% of this total and far exceed the contribution of any other ice shelf or region (Supplementary Fig. 1). Excluding these two ice shelves, EAIS-wide SGL volume peaked at 620 ×10^6 m^3 in January 2020 (Supplementary Fig. 1). SGL area and volume are strongly correlated (Supplementary Fig. 2) and, across the entire EAIS, we find that SGL area and volume anomalies fluctuate interannually by up to ~72% and ~61% (Fig. 2a–b). This variability is comparable to interannual meltwater volume variability on the Antarctic Peninsula (84%)[35] and on King Oscar Glacier in northwest Greenland[35] and Russell Glacier in west Greenland (64–83%)[36].

Interannual fluctuations in SGL volume are even higher on individual ice shelves, with the largest SGL volume anomalies in January 2020 on Moscow University Ice Shelf (225%), Riiser-Larsen (193%), and Shackleton (111%) ice shelves (Fig. 2c, g, i). Moreover, our results indicate peak years of SGL volume are asynchronous between ice shelves around the EAIS, including those experiencing similar mean annual surface melt near their grounding lines[37] (Fig. 2c–j; Table 1, Supplementary Figs. 3–5). For example, SGL volume peaked in 2019 on the Nivlisen (144 ×10^6 m^3) and Amery ice shelves (1731 ×10^6 m^3), but in January 2017 on the Roi Baudouin Ice Shelf (532 ×10^6 m^3), despite moderate mean annual surface melt rates derived from QuikS-CAT radar backscatter[37] of ~50–60 mm w.e. yr^−1 on all three ice shelves (Fig. 2d–f, Table 1). Years of peak SGL volume in 2017 and minimum SGL volume in 2019 on the Roi Baudouin Ice Shelf (Fig. 2) are consistent with years of maximum and minimum meltwater and slush extents derived from supervised classification of Landsat 8 imagery[38]. Conversely, SGL volume peaked in January 2020 (>77 ×10^6 m^3) on some of the northernmost ice shelves experiencing the most intense mean annual surface melt (>80 mm w.e. yr^−1), such as the Shackleton and Moscow University ice shelves (Fig. 2g, i, Table 1). In contrast, SGL volume peaked in 2014 in the regions experiencing the lowest mean annual surface melt (<20 mm w.e. yr^−1), such as the Nansen Ice Shelf in Victoria Land (60 ×10^6 m^3, Fig. 2j).

The large interannual variability in SGL volumes is also reflected in their spatial extent (Fig. 3). For example, we find evidence of SGLs spreading towards ice shelf calving fronts in successive melt seasons on the Nivlisen and Shackleton ice shelves and on to landfast sea ice (Supplementary Figs. 6 and 7). In addition, we find a weak correlation ($r = 0.35$, $r^2 = 0.13$, $p = 0.001$) between SGL areas and maximum SGL elevation. SGLs might be expected to reach higher elevations during melt seasons with more extensive surface meltwater, but our analysis suggests this is not the case (Supplementary Fig. 8). Indeed, the highest elevation at which SGLs occur inland varies substantially each year (Fig. 4). For example, maximum SGL elevation varies by over 1000 m interannually in Wilkes Land and Victoria Land, where lakes can form up to 1395 metres above sea level (m.a.s.l.) and 1895 m.a.s.l. respectively (Fig. 4). In the mountainous escarpment region of the Amery Ice Shelf, maximum SGL elevation varies by over 800 metres (Fig. 4). Across all ice shelves, the proportion of SGLs located above the grounding line varies interannually by up to 9% and we note that mean individual SGL area increases in years with more extensive SGL coverage (Table 2).

**Climatic influences on SGL variability.** To investigate the potential climatic drivers of the complex patterns in SGL variability, we explored the relationship between SGL areas and volumes with January and December-January-February (DJF) means of three climatic variables from ERA5 climate reanalysis: near-surface 2-m air temperature ($T_{2m}$), skin temperature ($T_s$), and net surface solar radiation ($S_{rad,}$ which represents the difference between downward incoming shortwave radiation and reflected longwave radiation[39]) (Table 1), as well as the preceding winter mean (February-December) snowfall, mean January snowfall and mean DJF snowfall (Supplementary Tables 1 and 4, Supplementary Fig. 9, Methods). These variables were chosen primarily because they have been identified as key controls on SGL development in Greenland and in Antarctica[26,40–42].

We find positive correlations between mean DJF $T_{2m}$ and SGL area and volume on all individual ice shelves, with the exception of the Roi Baudouin ice shelf (Fig. 5, Supplementary Table 4). This implies that more extensive, deeper SGLs form in warmer melt seasons when there is more melting, in agreement with

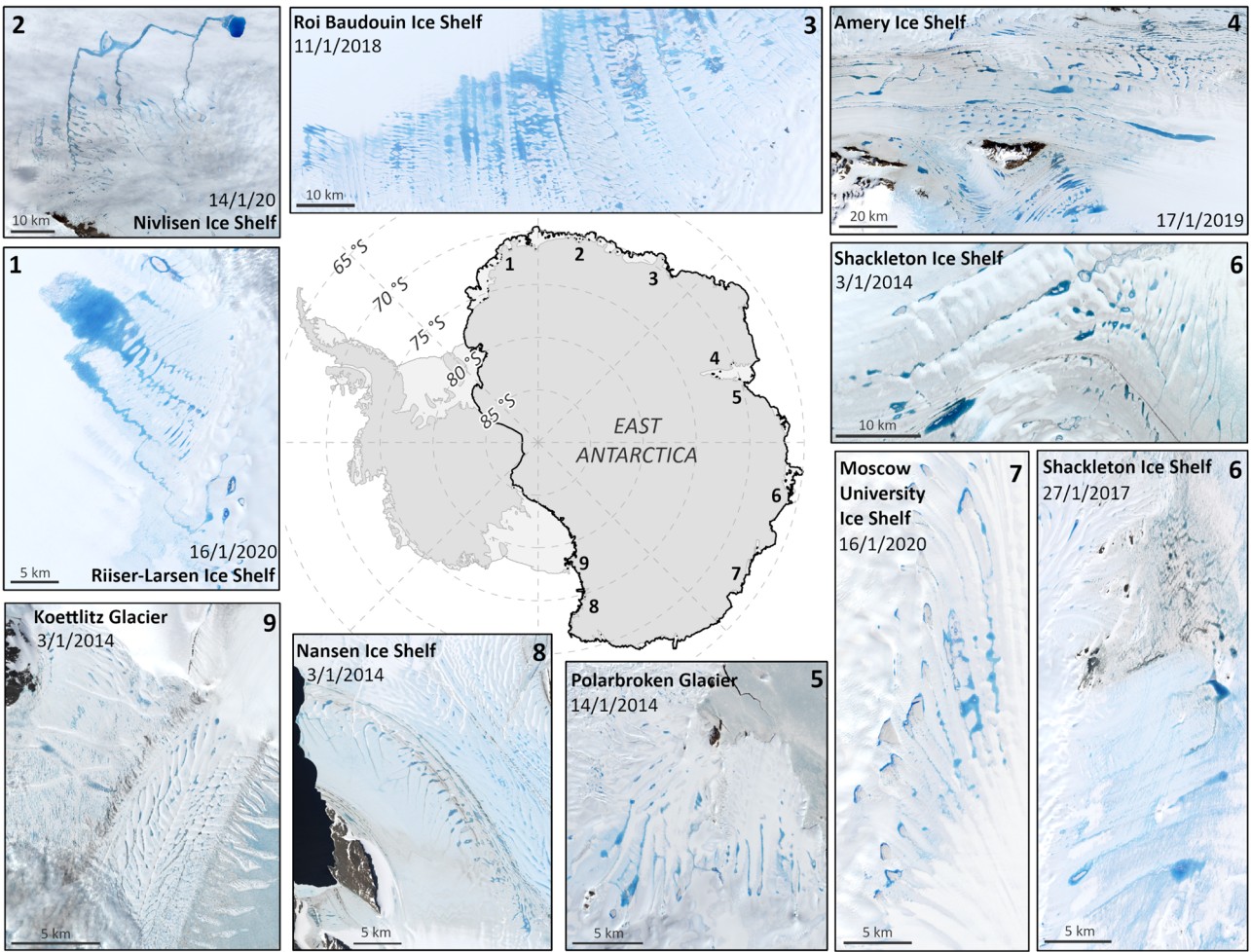

**Fig. 1 Examples of supraglacial lakes on selected East Antarctic ice shelves and outlet glaciers.** (1) Riiser-Larsen Ice Shelf, (2) Nivlisen Ice Shelf, (3) Roi Baudouin Ice Shelf, (4) Amery Ice Shelf, (5) Polarbroken Glacier/Publications Ice Shelf, (6) Shackleton Ice Shelf, (7) Moscow University Ice Shelf, (8) Nansen Ice Shelf and (9) Koettlitz Glacier. This figure highlights the distribution of SGLs across these major ice shelves but note that lakes occur with less frequency in other regions, for example along the Ingrid Christensen Coast (between Polarbroken Glacier and Shackleton Ice Shelf) and on Voyeykov Ice Shelf (adjacent to Moscow University Ice Shelf). Details of Landsat 8 images are in Supplementary Table 5. Grounding line from ref. [77] and coastline from ref. [78].

previous observations in East Antarctica[26,29,31] and on the Antarctic Peninsula[43]. The two regions of the ice sheet where DJF $T_{2m}$ is most strongly positively correlated with total SGL volumes are the Riiser-Larsen ($r = 0.67$, $p = 0.03$) and West ice shelves ($r = 0.79$, $p = 0.01$) (Fig. 5a, c). The negative correlation between mean DJF $T_{2m}$ and total SGL volume on the Roi Baudouin Ice Shelf suggests that interannual SGL variability on this ice shelf may be more influenced by other factors, such as the capacity of the firn to store meltwater[44]. We also investigated whether these correlations exist for data aggregated across the whole ice sheet (i.e. by performing linear regressions between mean DJF $T_{2m}$ and total SGL areas and volumes for all ice shelves together), but instead found a strong negative correlation between mean DJF $T_{2m}$ with total SGL area ($r = -0.72$, $p < 0.001$) and volume ($r = -0.68$, $p < 0.001$) and no significant correlation excluding the Amery and the Roi Baudouin ice shelves (Supplementary Table 1, Methods). This is likely to be because ice sheet-wide SGL areas and volumes are strongly skewed by the large SGL areas and volumes that form far from the coastline on the Roi Baudouin and Amery ice shelves, which have the two largest SGL area and volume contributions and the coldest grounding zone summer air temperatures on the EAIS (Table 1, Supplementary Fig. 1).

The preceding winter snowfall might be expected to be closely linked to SGL volumes because it provides firn air storage capacity for summer melt[16,18,44,45]. This means more snowfall during the preceding winter replenishes firn pore space, so surface meltwater can percolate into the firn and be stored rather than forming as SGLs[44]. However, we find low correlations between both total SGL area and volume with the mean snowfall during the preceding winter and with the mean summer snowfall (Supplementary Table 1, Fig. 5e–h).

Mean summer net surface solar radiation ($S_{rad}$) might be expected to be positively correlated to SGL volumes because enhanced incoming shortwave radiation (and therefore net $S_{rad}$) increases melting of the snowpack and the volume of liquid meltwater available to form SGLs[22,39]. However, we find low and negative correlations between both total SGL area and volume with mean DJF $S_{rad}$ ($r \leq -0.31$, $p \leq 0.05$, Supplementary Table 1, Fig. 5i–l). We attribute this to the limited influence of snowfall events and cloudiness on shortwave radiation transmissivity in ice shelf grounding zones compared to on coastal ice shelves[39]. Most SGLs form around or just downstream of low-albedo blue (bare) ice regions close to ice shelf grounding lines, where snow and firn have been entirely removed by persistent katabatic wind erosion, sublimation or melt[15,26,29]. Here, there is regular melting because

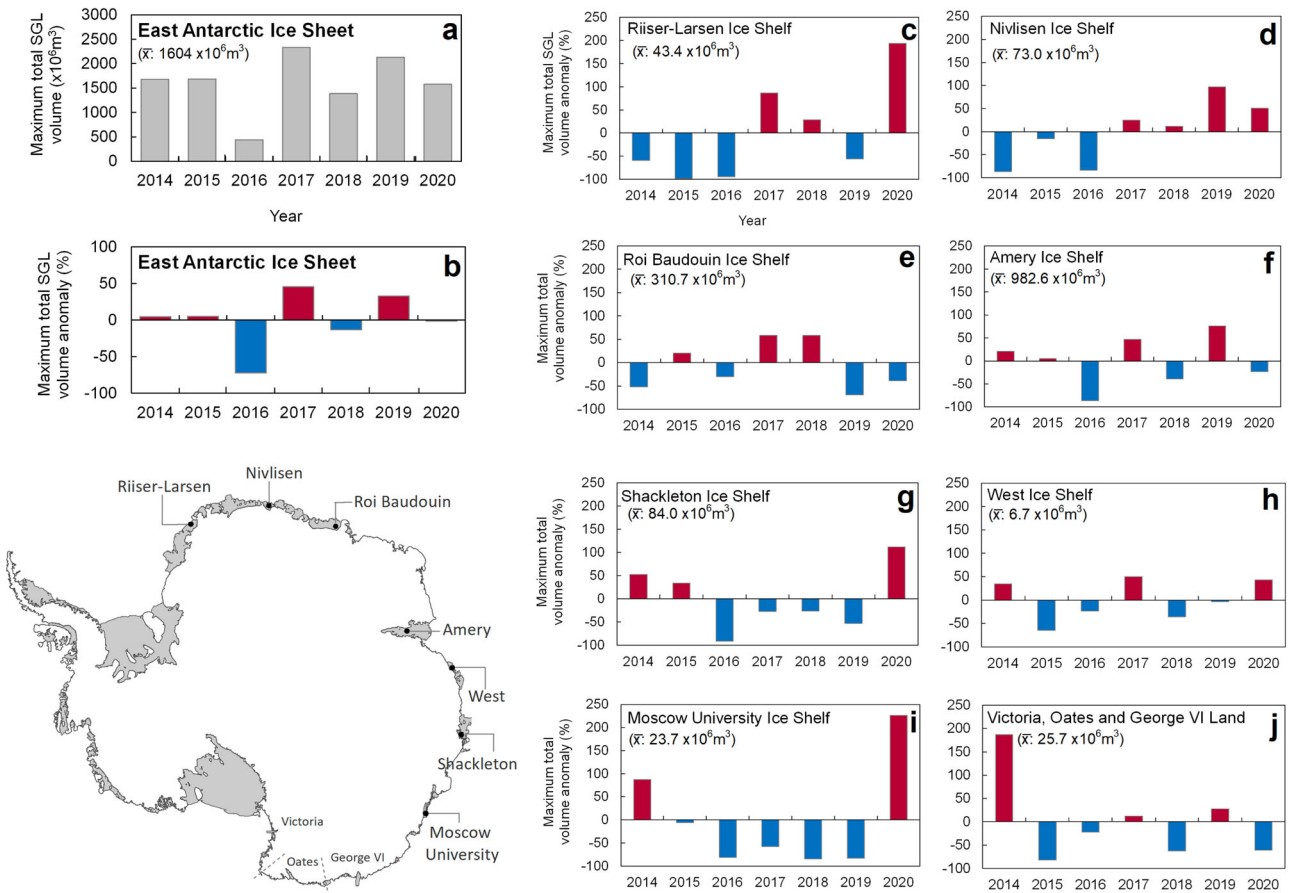

**Fig. 2 Interannual changes in supraglacial lake volumes on the East Antarctic Ice Sheet. a** Absolute total SGL volumes (in millions of cubic metres) on the East Antarctic Ice Sheet. **b** Percentage SGL volume anomalies (i.e. percentages of the mean 2014–2020 maximum total lake volume) on the East Antarctic Ice Sheet. **c–j** Percentage SGL volume anomalies (i.e. percentages of the mean 2014–2020 maximum total lake volume) on selected major ice shelves and regions. Positive anomalies are shown in red and negative anomalies are shown in blue. See Supplementary Fig. 3 for anomalies as standard deviations. The absolute mean maximum total SGL volume ($\bar{x}$) is shown in panels **a** and **c–j**. Supraglacial lake volume anomalies for two addition regions, the Ingrid Christensen Coast and Voyeykov Ice Shelf, are shown in Supplementary Fig. 4 rather than this figure because lakes occur with less frequency in these two regions. Grounding line from ref. [77] and coastline from ref. [78].

**Table 1 Mean 2014–2020 values of climatic variables extracted from ice shelf/outlet glacier grounding zones (Methods, Supplementary Fig. 14).**

| Ice Shelf/Glacier | Mean January 2014–2020 maximum total SGL volume ($\times 10^6$ m$^3$) | $T_{2m}$ (°C) | $T_s$ (°C) | $S_{rad}$ (W m$^2$) | Snowfall ($\times 10^{-5}$ m w.e.) | MAR Mean annual 1979–2020 melt (mm w.e. yr$^{-1}$) | Mean annual 1999–2009 melt (mm w.e. yr$^{-1}$)[37] |
|---|---|---|---|---|---|---|---|
| Nivlisen | 73.0 | −5.3 | −7.6 | 0.1 | 4.2 | 15.2 | 55.1 |
| Shackleton | 84.0 | −7.7 | −7.4 | 2.2 | 4.7 | 49.6 | 115.1 |
| Roi Baudouin | 310.7 | −9.7 | −10.5 | 0.2 | 3.4 | 53.1 | 61.3 |
| Riiser-Larsen | 43.4 | −8.4 | −10.2 | 0.3 | 3.9 | 16.6 | 42.1 |
| Moscow University | 23.7 | −5.1 | −4.8 | 3.5 | 7.6 | 22.3 | 98.6 |
| Voyeykov | 16.2 | −3.4 | −1.9 | 2.0 | 8.7 | 6.0 | 85.3 |
| Amery | 982.6 | −12.4 | −13.0 | 1.1 | 1.7 | 15.6 | 49.8 |
| Nansen | 25.7 | −7.4 | −6.8 | 4.4 | 1.4 | 5.0 | 13.3 |
| West | 6.7 | −5.6 | −5.2 | 2.0 | 5.3 | 19.6 | 90.6 |
| Publications | 35.8 | −6.8 | −7.9 | 1.1 | 3.8 | 26.3 | 68.7 |
| Matusevitch | 1.5 | −5.1 | −3.8 | 4.8 | 5.7 | 13.7 | 73.7 |
| Skelton | 13.3 | −5.2 | −4.8 | 5.8 | 1.0 | 0.59 | 17.3 |

2-m temperature ($T_{2m}$), skin temperature ($T_s$), surface net solar radiation ($S_{rad}$) and mean winter (February to December) 2014–2020 snowfall are simulated by ERA5 reanalysis. $T_{2m}$, $T_s$ and $S_{rad}$ are mean January values. Mean annual 1979–2020 surface melt simulated by MAR (this study) and mean annual 1999–2009 surface melt fluxes derived using an empirical relationship between QuikSCAT satellite scatterometer observations between 1999–2009 and melt calculated from in situ energy balance observations[37]. These scatterometer-derived melt fluxes are provided for reference and comparison with MAR-derived mean annual melt estimates, but are not the melt rates used in our statistical analysis or in the firn model simulations.

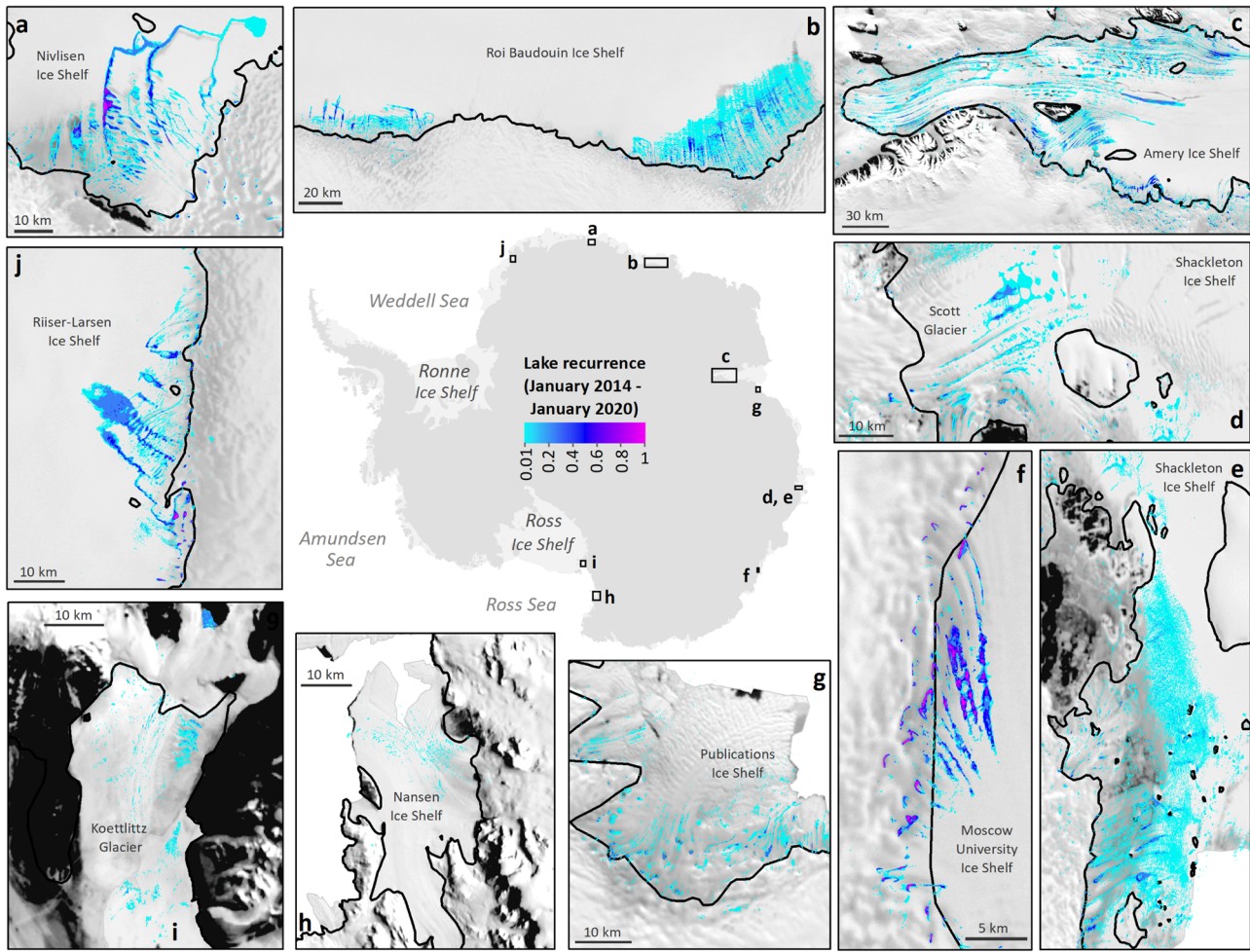

**Fig. 3 Supraglacial lake recurrence around East Antarctica. (a–j)** Normalised count of overlapping lakes at their maximum extent during January from 2014 to 2020 on selected major ice shelves and outlet glaciers, weighted according to the number of useable (partially or totally cloud-free) satellite images in this period. Turquoise/pale blue colours correspond to infrequently-forming lakes (i.e. that formed in a single year) and pink/purple colours correspond to frequently-forming lakes (i.e. that formed on multiple dates in January in several or all years). This figure highlights the distribution of SGLs across these major ice shelves but note that lakes occur with less frequency in other regions, for example along the Ingrid Christensen Coast (between Polarbroken Glacier and Shackleton Ice Shelf) and on Voyeykov Ice Shelf (adjacent to Moscow University Ice Shelf). The grounding line is shown as a solid black line in all panels. Grounding line from ref. [77] and coastline from ref. [78].

the reduced albedo increases shortwave radiation absorption, and katabatic winds maintain enhanced turbulent surface heating[29,31]. This enables SGLs to form at high elevations where low-albedo exposed bedrock and blue ice are abundant[15,23] (Fig. 4). High summer surface melt rates near ice shelf grounding lines counteract the increase in FAC caused by snowfall events during the preceding winter[42]. Therefore, these weak correlations indicate seasonal snowfall and $S_{rad}$ fluctuations may not be the primary drivers of surface melt availability, and hence interannual variability in SGL area and volume.

The only region where we find a strong, significant positive correlation between $S_{rad}$ and SGL volumes is on the Amery Ice Shelf ($r = 0.84$, $p = 0.01$, Fig. 5k, Supplementary Table 4). Despite recording the largest SGL volumes, Amery experiences low near-surface summer temperatures for its latitude and the lowest $T_{2m}$ and $T_s$ on the EAIS (Table 1), with a grounding zone extending far south, meaning it experiences a relatively cold climate in that location (Fig. 5c). This means intermittent surface melt makes SGL variability sensitive to albedo variations controlled by snowmelt-albedo feedbacks, where snow containing refrozen meltwater and where blue ice exposed by katabatic winds have a lower surface albedo than fresh snow and therefore absorb more

incoming solar radiation, leading to more surface melt[22,46]. Increased incoming shortwave radiation has been linked to more extensive SGLs on the Amery Ice Shelf[31]. This suggests interannual variability in SGL volume on Amery is largely driven by the absorption of shortwave radiation, rather than variability in near-surface summer air temperatures or snowfall.

**Influence of surface conditions on SGL variability.** Depleted FAC, refrozen subsurface meltwater and high surface runoff volumes have been linked to SGL formation on East Antarctic ice shelves[22,27,45] and on the Antarctic Peninsula[17,47] and can render ice shelves more susceptible to hydrofracture[14]. Meltwater percolating through the firn that encounters shallower impermeable ice lenses (formed by refrozen meltwater) can percolate less far into the ice before fully saturating the snowpack[17,40,48]. Continued surface melting over successive melt seasons gradually depletes FAC when more pore space is lost by melt and refreezing during densification than is replenished by snowfall[44]. At this point, meltwater can no longer percolate into and refreeze within the firn, meaning the firn cannot act as an absorbing buffer[16,44]. Meltwater percolation and refreezing into the firn layer can also

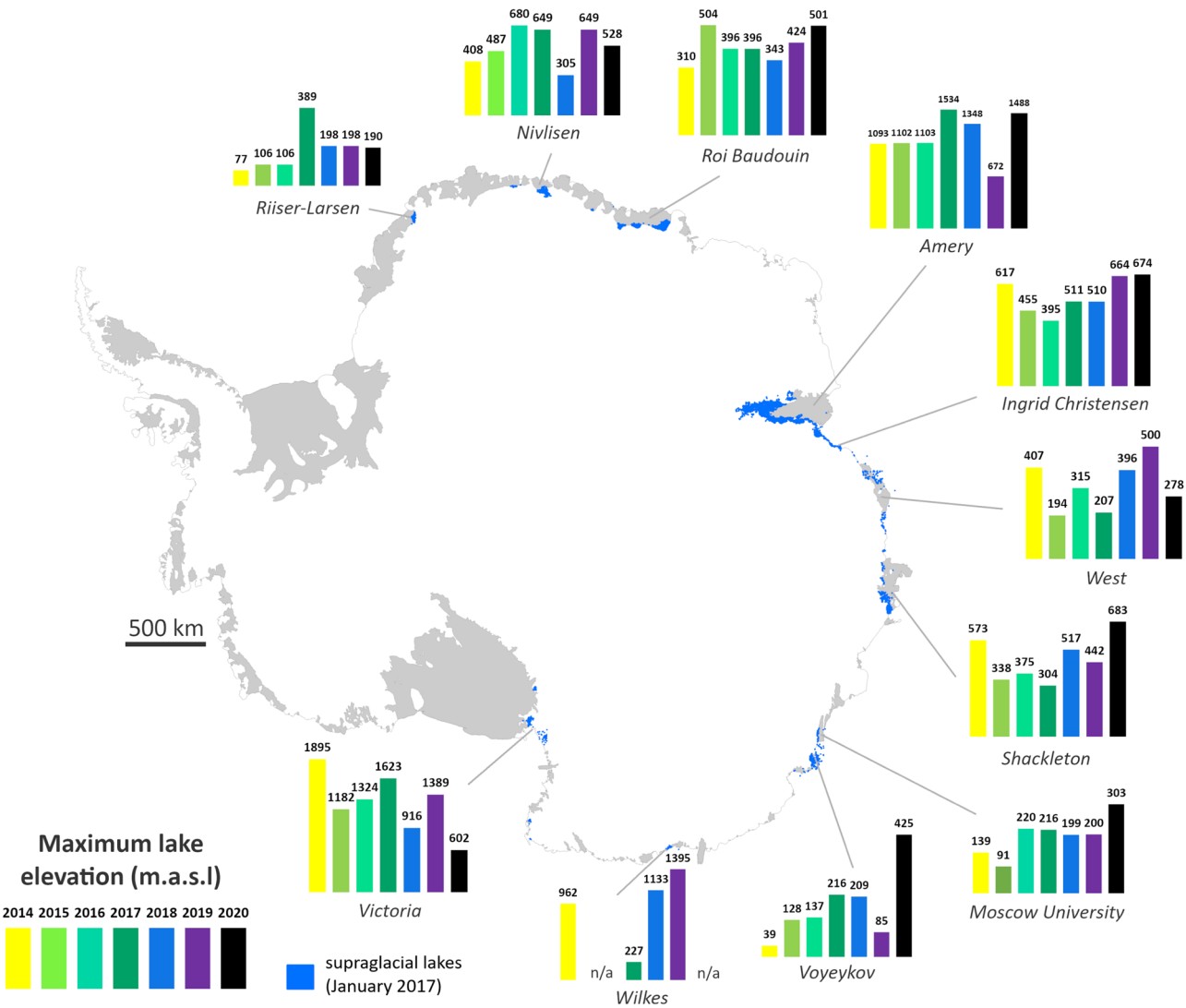

**Fig. 4 Maximum elevation of supraglacial lakes in January from 2014 to 2020 around East Antarctica.** Key ice shelves/regions are highlighted. Grey areas are ice shelves and floating glacier tongues. Grounding line from ref. [77] and coastline from ref. [78]. Lake extents in January 2017, the most extensive lake year, are shown in blue.

**Table 2 Summary of annual January SGL metrics.**

|                                          | 2014  | 2015  | 2016  | 2017  | 2018  | 2019  | 2020  |
|------------------------------------------|-------|-------|-------|-------|-------|-------|-------|
| Total SGL-covered area (km$^2$)          | 2080  | 2237  | 757   | 2512  | 1835  | 2236  | 2186  |
| Proportion of SGLs by area on floating ice (%) | 76 | 85 | 75 | 84 | 78 | 76 | 80 |
| Total SGL volume (×10$^6$ m$^3$)         | 1679  | 1684  | 438   | 2332  | 1386  | 2130  | 1577  |
| Mean individual SGL area (km$^2$)        | 0.034 | 0.035 | 0.022 | 0.048 | 0.033 | 0.035 | 0.037 |
| Maximum SGL elevation (m a.s.l)          | 1895  | 1182  | 1324  | 1623  | 1348  | 1395  | 1488  |

Total SGL-covered area represents the sum of the maximum area covered by individual SGLs in January. Total SGL volume represents the sum of the maximum SGL volume mask (Methods).

exert a localised warming effect on ice temperatures through the release of latent heat[16,17]. When firn saturation prevents meltwater percolation and refreezing within the firn, the firn is flooded and excess surface runoff can form SGLs[18,38,49]. To investigate these potential controls on the interannual variability in SGL area and volume, we performed linear regressions between total SGL areas and volumes and mean November FAC, mean summer (January and DJF) FAC, mean summer total runoff and mean summer shallowest ice lens depth simulated by the CFM[32]. We also conducted linear regressions with mean summer surface

melt simulated by the regional climate model MARv3.11[33], as well as the November FAC-to-DJF surface melt ratio (Methods). November FAC reflects accumulation throughout the year and hence the firn meltwater storage capacity before the onset of summer melt. We therefore included November FAC in our regression to try and separate the effects of summer melt and firn storage capacity on interannual SGL variability. We also compared mean January FAC and DJF FAC with total SGL areas and volumes to assess the influence of FAC around the peak of the melt season[2,15,26,28,29] and over the whole melt season.

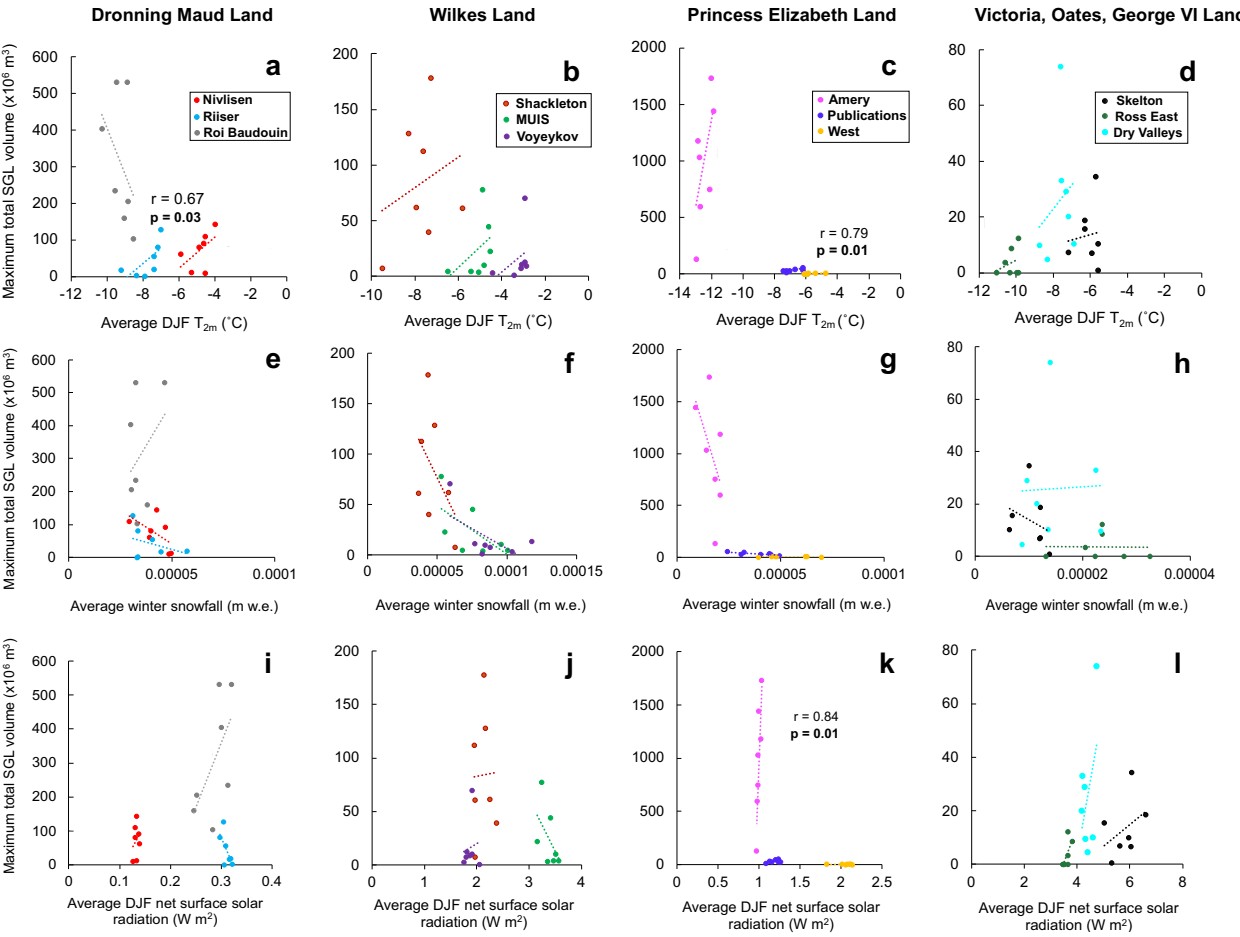

**Fig. 5 Relationships between climatic variables and supraglacial lake volumes on the East Antarctic Ice Sheet.** Scatter plots of mean December-January-February (DJF) 2-m temperature ($T_{2m}$) (**a–d**) mean snowfall in the preceding winter (February to December) (**e–h**) and mean DJF net surface solar radiation ($S_{rad}$) (**i–l**) from ice shelf grounding zones simulated by ERA5 reanalysis (see Methods) and maximum total SGL volume grouped by major EAIS region. Individual ice shelves are represented by different colours (see Fig. 1 for locations). Significant relationships ($p < 0.05$) in a linear regression are displayed.

Across the whole ice sheet, total SGL volume is most strongly correlated with mean summer FAC ($r = -0.37$, $p < 0.001$) (Supplementary Table 2). This correlation is even higher for total SGL area ($r = -0.44$, $p < 0.001$) (Supplementary Table 2). SGL volume is more weakly but significantly correlated with mean summer minimum ice lens depth ($r = -0.24$, $p < 0.05$) and is not correlated with mean summer surface melt or the November FAC-to-DJF surface melt ratio ($r \geq 0.16$, $p > 0.05$, Supplementary Table 2). Therefore, the firn meltwater storage capacity is likely to exert greater influences on total SGL volume than mean summer melt or ice lens depth. This relationship is closely linked to the presence of saturated firn, which controls the amount of snowmelt that can percolate downward into the snowpack and be absorbed in the firn layer.

Perhaps surprisingly, there exist substantial discrepancies between observed total SGL volumes and surface runoff modelled by CFM-MAR (Fig. 6i–l). The only two locations where we find SGL volumes are significantly positively correlated with modelled surface runoff are the Shackleton ($r = 0.86$, $p = 0.01$) and Riiser-Larsen ($r = 0.76$, $p = 0.04$) ice shelves (Fig. 6i, j, Supplementary Table 3). For these two cases, we use CFM-MAR to further investigate the potential influences on SGL development and find January surface melt and November FAC-to-DJF melt ratio are important climatic predictors of interannual SGL variability. On the Shackleton Ice Shelf, higher SGL volumes correlate with

higher mean January surface melt ($r = 0.73$, $p = 0.05$) and a lower November FAC-to-DJF melt ratio ($r = -0.65$, $p = 0.02$, Fig. 6b, f). On the Riiser-Larsen Ice Shelf, we find SGL volumes are also strongly positively correlated with mean January surface melt ($r = 0.75$, $p = 0.04$, Fig. 6e). This is in good agreement with high mean annual melt days on these two ice shelves (>78 annual melt days in places) observed from passive microwave data between 1979–2020[50]. Weaker, non-significant correlations exist between total SGL volume and mean January minimum ice lens depth on both ice shelves ($r \leq -0.57$, Fig. 6m–n). These relationships between SGL volumes, January surface melt and November FAC-to-DJF melt ratio are consistent with higher melt-to-snowfall ratios depleting FAC in the grounding zones of these ice shelves, resulting in excess surface meltwater and large seasonal SGL volumes[18,29,44].

Of potential significance is that the mean January 2020 modelled surface melt rate on the Shackleton Ice Shelf (0.51 m w.e. a$^{-1}$, Fig. 6j) approached the previously-suggested upper melt rate threshold for Antarctic ice-shelf viability of 0.725 m w.e. a$^{-1}$ (ref. [12]). Previous work has suggested its vulnerability to hydrofracturing is currently low because only a small portion of SGLs form in high-tensile regions that provide buttressing[22]. However, we record very high total SGL area and volume across Shackleton Ice Shelf in January 2020 compared to previous years (e.g. 156% higher than in January 2017). Although this high

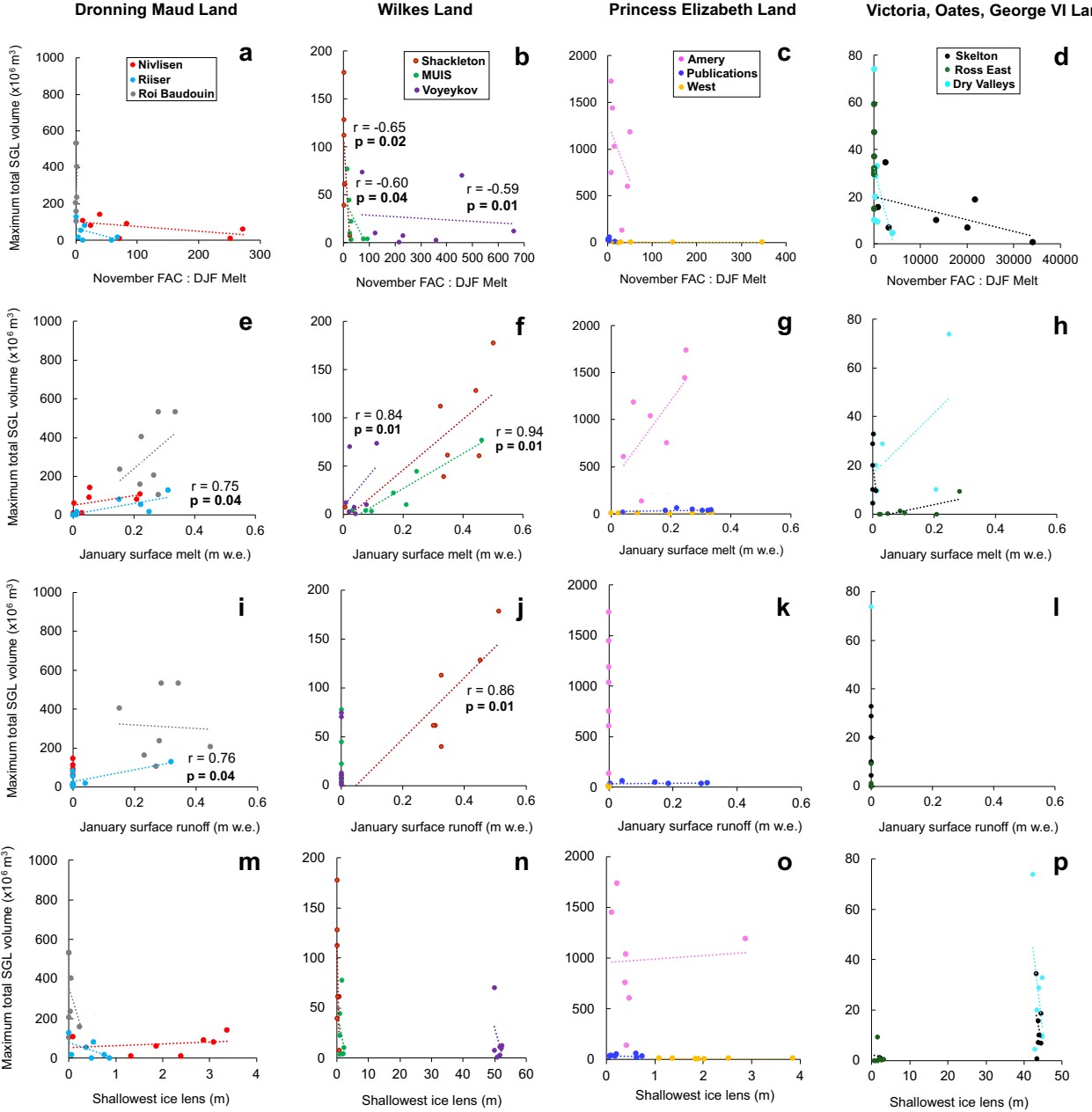

**Fig. 6 Relationships between near-surface conditions and supraglacial lake volumes on the East Antarctic Ice Sheet.** Scatter plots of mean November firn air content-to-DJF melt (**a–d**), mean January surface runoff (**i–l**) and depth of shallowest ice lens (**m–p**) (simulated by the Community Firn Model) and mean January total surface melt (**e–h**) (simulated by MAR) against maximum total SGL volume, grouped by major EAIS region. Individual ice shelves are represented by different colours (see Fig. 1 for locations). Significant relationships ($p < 0.05$) in a linear regression are displayed.

annual variability and our relatively short time series do not confirm an increasing trend, we observe SGLs already intersected regions vulnerable to hydrofracture in 2014, 2015 and 2020 (Supplementary Figs. 6 and 10). Moreover, Shackleton Ice Shelf experiences a high number of mean annual melt days (68 annual melt days on parts of the ice shelf) and one of the earliest melt season onsets in East Antarctica, typically starting at the end of November[50]. Thus, this ice shelf has been highlighted as being susceptible to future hydrofracturing, given that current mean annual melt in places is already close to the 71 days per year annual mean runoff projected at 4 °C of warming above pre-industrial levels[14]. Should this occur, SGLs are likely to become deeper and more extensive as summer surface melting increases with atmospheric warming under higher-emissions scenarios[11,12].

We suggest, therefore, that surface meltwater delivery to regions vulnerable to hydrofracture could bring ice shelves like Shackleton closer to the threshold of instability and hydrofracture-driven collapse.

Despite the strong relationships on some ice shelves, the relationship between SGL volume and modelled surface runoff, melt, ice lens depth and November FAC-to-summer melt ratio weakens on ice shelves with complex surface topography and/or steep coastal escarpment regions, such as in Dronning Maud Land[46] (Fig. 6). For example, there are low individual correlations ($r < 0.46$) between SGL volumes and FAC-to-melt ratio, surface melt, runoff and minimum ice lens depth on the Roi Baudouin and Nivlisen ice shelves (Fig. 6a, e, i, m, Supplementary Table 3). We suggest this may be related to high snow erosion rates and

intense surface melt within persistent blue-ice areas in the grounding zones of both ice shelves, driven by katabatic winds descending steep ice surface slopes[22,28,38]. Amery Ice Shelf is another topographically-complex region where SGL volumes show low correlations with modelled FAC-to-melt ratio ($r = -0.42$, $p = 0.22$) and minimum ice lens depth ($r = 0.05$, $p = 0.89$), despite a stronger relationship with surface melt ($r = 0.70$, $p = 0.10$, Fig. 6c, g, k, o, Supplementary Table 3). The low correspondence between simulated FAC-to-melt ratio, surface melt, and SGL volume in these regions may reflect the regional climate model spatial resolution, which smooths the topography and likely produces biases in temperature, snowfall and FAC build-up[51]. Consequently, localised controls on SGL formation and evolution through the melt season are not fully resolved, particularly melt-albedo feedbacks[15,22,23,29]. Katabatic wind-driven snow scouring and surface melting is a crucial control on SGL formation, as are exposed rock outcrops, which locally increase wind speeds and surface melting[15,23]. Furthermore, the representation of liquid water processes and the ability of firn to accommodate melt refreezing is a weakness of firn models, meaning they may underestimate runoff that is available to pond as SGLs[52].

Finally, we note that mean January modelled surface runoff is zero in most regions of the ice sheet where SGLs form, despite mean January 2014–2020 maximum total SGL volumes of up to $982.6 \times 10^6$ m$^3$ (Fig. 6i–l, Table 1). This suggests MAR may be potentially overestimating annual accumulation, which may be related to drifting-snow transport processes not included in MAR[53]. This would increase firn porosity and its capacity to store and refreeze surface meltwater[16,17,44], resulting in surface runoff underestimates. In addition, high-magnitude, low-frequency snow accumulation events linked to atmospheric rivers could be an important influence on the asynchronicity of SGL volume between different ice shelves (e.g. Fig. 2d–f). Such events can lead to extensive surface melt in regions that rarely experience melting[54,55] and can cause large interannual variations in snowfall[53]. We recommend that future model development focus on the implementation of local-scale wind-driven ablation over the EAIS to better resolve the spatial variability in snow accumulation processes[56]. This will be important for improving agreement between observed SGL volumes and surface runoff estimates in topographically-complex ice shelf regions where melt-albedo feedbacks dominate[15,22,23].

**Summary**. We have produced the first observations of SGL area and volume around the peak of seven consecutive melt seasons from 2014 to 2020 around the entire EAIS. Our results demonstrate that SGL volume varies interannually by as much as 200% ($>2\sigma$) on individual ice shelves and by up to ~72% across the entire ice sheet. Peak years of SGL volume are asynchronous between ice shelves, including those experiencing similar mean annual surface melt. SGL area and volume are strongly positively correlated with mean summer (December-January-February, DJF) air temperature on almost all individual ice shelves. Interannual SGL variability is also sensitive to mean summer firn air content (FAC). We find substantial discrepancies between observed total SGL volumes and surface runoff modelled by MAR. However, on the few ice shelves where these are significantly correlated, January surface melt and the ratio of November FAC to DJF melt are important predictors of SGL volume. This suggests potentially large increases in SGL coverage and volume should be expected under increased atmospheric warming, meaning SGLs are likely to spread to ice-shelf regions vulnerable to hydrofracture[6,20]. These results highlight the important interactions between local climate and ice/firn surface

properties in controlling the interannual variability in SGLs around the EAIS, which are complex but nonetheless require a better understanding for improved predictions of surface meltwater ponding in Antarctica.

## Methods

**East Antarctic Ice Sheet-wide mapping of SGL extents**. We applied the threshold-based pixel classification method[34] to all available Landsat 8 imagery (30 m resolution) from 2014 to 2020. To obtain a record of SGLs around the peak of the melt season, we selected images from January of each year, which typically coincides with the peak of the austral summer around marginal regions of East Antarctica[2,15,26,28,29], that were acquired at sun elevation angles greater than 20°. We used imagery from the month of January because our aim is to produce a comprehensive, consistent comparison of maximum total surface meltwater around the EAIS over this seven-year period, rather than focusing on its temporal evolution over the entire melt season. Quantifying this interannual variability in surface meltwater is an important step in advancing process understanding of which ice shelves may be closer to potential thresholds of meltwater-induced hydrofracturing. Our work also extends previous work[15] which provided the first ice sheet-wide assessment of supraglacial lake areas at the peak of the melt season for a single melt season (January 2017). Though late-onset melting on some ice shelves in some years may result in surface runoff peaking later in the melt season, we find lake areas and volumes from Landsat 8 imagery typically peak between mid-late January during our study period, after which lakes typically start freezing over. Including imagery from February is therefore unlikely to have substantially changed our maximum lake area and volume estimates. Furthermore, regional climate models have been shown to simulate peak surface runoff in January between 2005 and 2020 over the Amery Ice Shelf[57] and several other studies also note that maximum meltwater extent occurs in mid-late January[2,15,26,28,29]. Including the rest of the melt season (December and February) in our analysis would also have been significantly more computationally expensive. We focus here on Landsat 8 (launched in February 2013) rather than including imagery from Landsat 7 or earlier sensors because the algorithm we use was fine-tuned using a training dataset of 15 Landsat 8 images covering a wide range of illumination conditions, cloud cover, geology, and spectral characteristics[34]. In total, we processed 2175 Landsat 8 images. On each ice shelf or coastal region where SGLs form, useable images (i.e. those not completely obscured by cloud cover) covered on average 24 days in January from 2014 to 2020, and was highest on Amery Ice Shelf (29 days) and lowest in Wilkes Land (9 days) (Supplementary Figs. 11a and 12). The number of total useable images in January in any given year varied from 6–10 in some regions (e.g. Amery Ice Shelf, Dry Valleys) to 1–3 in others (e.g. Wilkes Land, Supplementary Fig. 11b–q). We focus exclusively on the EAIS because surface meltwater is widespread around its margins[15,23,30] and forms on ice shelves potentially vulnerable to hydrofracturing[18,22].

The pixel-based classification combines separate threshold-based algorithms to detect (1) surface meltwater, (2) clouds, (3) exposed rock outcrop and (4) seawater. The full details are discussed comprehensively in ref. [34], but are briefly outlined here. Liquid water-covered pixels are classified using the Normalized Difference Water Index[29,30]. Rock, seawater and cloud are classified using the Normalized Difference Snow Index[58,59] and the Thermal Infrared Sensor (Band 10)/Blue (Band 2) ratio. Further thresholds are applied to exclude cloud shadows and shaded snow areas. Threshold values were determined by creating a training dataset based on selected Landsat 8 images. Using these thresholds, binary (i.e. meltwater and non-meltwater) masks are created for each Landsat 8 scene. Areas <5 pixels (30 m resolution) in total and linear features that are narrower than 2 pixels are removed from these masks to avoid ambiguous classification of mixed slush as surface meltwater[15,27,60].

We used the GDAL (Geospatial Data Abstraction Library) 'Polygonize' utility to create vector polygons (shapefiles) of each masked Landsat 8 scene, where regions sharing a common pixel value (i.e. '1' = SGL, '2' = rock/seawater, '3' = cloud) are assigned this value as an attribute. For January of each year, we extracted and combined all polygons classified as SGL in the Geographic Information Systems package ArcMap. We include a small number of SGLs forming on landfast sea ice attached to ice shelves, as this is a perennial feature around the ice sheet and experiences surface melt ponding[61,62]. To quantify cloud cover for January of each year around the EAIS margin, we extracted and combined all polygons classified as cloud and counted overlapping features in ArcGIS Pro (Supplementary Fig. 13). This enabled us to confirm that low SGL occurrence was not simply an artefact of a low number of useable images.

In the absence of in-situ validation data, we manually verified our classification results for all 2175 Landsat 8 images and removed any false positives (cloud, shadow or rock mis-identified as SGLs that bypassed initial cloud, rock and seawater masking procedures due to spectral similarities). These false positives were often distinguishable by their 'diffuse' boundaries, as opposed to distinct lake objects. False positives tended to be minimal (typically < 1% of total individual SGLs) on large ice shelves such as Amery, Roi Baudouin and Riiser-Larsen, whereas they were much higher (up to 95%) in high-elevation regions with exposed nunataks and dirty ice, in particular Victoria and Oates Land. Therefore, manual post-processing was necessary to avoid over-prediction of SGL occurrence in these

regions. ref. [34] recorded an accuracy of >94% when validating SGLs classified using our method against manually-digitised SGLs. Finally, we created maximum lake area masks for January of each year (i.e. containing pixels that were classified as lake on at least one day in January) by stacking and merging lake outlines for all dates within January for which we were able to classify lakes. We did this to be able to calculate maximum lake volume masks (below) due to temporally varying satellite paths and/or variable cloud cover around the EAIS margin[63]. We used the REMA (Reference Elevation Model of Antarctica[64]) mosaic (200-m resolution) to extract SGL elevations. To quantify SGL recurrence, we used the 'Count Overlapping Features' tool in ArcGIS Pro.

**East Antarctic Ice Sheet-wide generation of SGL volumes**. We calculated lake depths and volumes using a physically-based model that has been widely used in similar studies of SGLs in Greenland and Antarctica[27–29,34,65,66]. This model is based on the rate of light attenuation in water and makes a number of assumptions, including that the lake bottom has a homogenous albedo, that there is little to no particulate matter in the water column to alter its optical properties, and that there is minimal wind-induced surface roughness[66]. For January of each year (2014–2020), we created a maximum lake depth mask by assigning all water pixels in the maximum lake area mask a depth equal to the maximum water depth observed out of all images during January following ref. [63], using the Cell Statistics tool in ArcGIS. Spatiotemporally variable satellite image acquisition and/or variable cloud cover around the EAIS margin mean that, for different ice shelves, lake depths are calculated on different days in January (Supplementary Fig. 11), so this approach allows us to create a single lake depth mask for January of each year per ice shelf/glacier. We then clipped the resulting lake depth mask to the extent of the lake area mask and multiplied each pixel by its area ($900\ m^2$, due to the 30 m resolution of Landsat 8) to create a final maximum lake volume mask.

**Comparison with climatic variables**. We conducted individual linear regressions of total lake area and volume with January and December-January-February (DJF) means of the following ERA5 reanalysis variables: 2-m air temperature ($T_{2m}$), skin temperature ($T_s$), and surface net solar radiation (amount of direct and diffuse incoming shortwave solar radiation minus the amount reflected by the Earth's surface), as well as mean preceding winter snowfall (i.e. total amount of accumulated snow during the February-December period preceding austral summer) (Table 2). We performed this correlation analysis to test the degree to which these climatic variables influence interannual variability in lake area and volume. ERA5 reanalysis is created by assimilating satellite and in situ observations since 1979 into the European Centre for Medium-Range Weather Forecast's (ECMWF) Integrated Forecast System, and is provided at a 0.25° (~31 km) horizontal resolution. For each variable, we calculated January and December-January-February (DJF) means from daily outputs for the period 2014–2020 within manually-delineated polygons of each ice shelf/outlet glacier grounding zone where SGLs form (Supplementary Fig. 14). ERA5 grid cells were excluded from calculations if their majority (i.e. mid-point) did not intersect with these polygons (Supplementary Fig. 15). This was to avoid means being skewed by ERA5 grid cells that only covered a very small portion of the ice shelf grounding zone, for example those located further inland from the grounding line.

We perform parametric correlation analysis because we wanted to test the degree to which different climatic and surface conditions influence interannual variability in total SGL area and volume. Furthermore, we wanted to assess whether these variables simulated by ERA5, the CFM and MAR (mean November FAC, mean summer FAC, mean summer total runoff, mean summer shallowest ice lens depth, near-surface 2-m air temperature, skin temperature, net surface solar radiation, mean preceding winter snowfall and mean summer snowfall) can be used as a first-order prediction of SGL area and volume over Antarctic ice shelves, as this is one of the first studies to make such comparisons with this range of variables. Several previous studies have performed Pearson correlation to assess potential drivers of interannual variability in observed total lake area, including with modelled seasonal snowmelt, surface air temperature, and firn air content[5,26,29,31,57]. The sample sizes in these preliminary correlation analyses are typically limited by the length of the satellite record, as in this study. We note that even if we had extended our observational period back in time to also include the Landsat 7 record (i.e. from 2000 to 2020 rather than from 2014 to 2020), which would require significant computational resources, the number of data points in each linear regression would still be <30.

**Comparison with firn model and regional climate model outputs**. We conducted individual linear regressions of total lake area and volume with the following variables simulated by the Community Firn Model (CFM[32]): mean January firn air content in the upper 10 m (FAC), mean DJF FAC, mean November FAC, mean January and DJF total runoff, mean January and DJF shallowest ice lens depth. We also conducted individual linear regressions of total lake area and volume with mean January and DJF surface melt simulated by the regional climate model MARv3.11 (hereafter MAR[33]), as well as the November FAC-to-DJF surface melt ratio. For each variable, monthly means of daily firn model outputs were

extracted from the closest MAR grid cell to 11 locations (Supplementary Fig. 11) within ice shelf/outlet glacier grounding zones where SGLs form.

The CFM uses the $Ar_{MAP}$ firn densification model[52,67]. $Ar_{MAP}$ is a parameterisation of firn densification equations, which have been calibrated in a Bayesian framework to an extensive dataset of 91 firn depth-density profiles, of which >60 are from Antarctica (>30 on the East Antarctic Ice Sheet) and showed good performance in capturing firn air content at these calibration sites[68]. Full details of $Ar_{MAP}$ are discussed in ref. [52]. In situ firn density measurements can be subject to measurement uncertainty of ~10%[52,67] which can translate into the CFM results. The CFM allows meltwater infiltration, retention and refreezing in the firn using a bucket scheme, where meltwater is allowed to percolate through the firn[52]. Meltwater is assumed to run off as liquid water that is available to form SGLs once it reaches the depth below which density remains greater or equal to 830 kg m[3] (i.e. the pore close-off depth). This means meltwater can bypass discrete ice lenses within the firn column, representative of firn evolution on a large horizontal scale. Volumetric irreducible water content (liquid water held by capillary forces) was set to 2% of the pore volume. Meltwater is refrozen in a model layer only if the layer has sufficient pore space, or else percolates down until it reaches a layer of maximum density. Each simulation consists of a spin-up by repeating a reference climate until reaching a firn column in equilibrium. The reference climate is taken as 1979–2009 with a spin-up duration of 120 years.

The CFM was forced with temperature, snow accumulation and melt rates from MAR[33]. MAR has been extensively used to study Antarctic surface mass balance and surface melt[14,69–72]. An extensive description of the MAR set-up for Antarctica can be found in ref. [33]. MAR is forced at its atmospheric lateral and upper boundaries with temperature from the ERA-Interim reanalysis from the ECMWF[73] and is run at a 35-km horizontal resolution. Compared to previous versions, MARv3.11 uses an improved ice-sheet mask based on Bedmap2[74] that includes rock outcrops, enabling potential enhanced melt-albedo feedbacks around exposed rocks[15,23]. Uncertainties in MAR are difficult to quantify and propagate from ERA-Interim, which has shown a warm bias of +3–6 °C over the Antarctic plateau[75]. The previous model version (MARv3.10) was found to underestimate surface mass balance over ice shelves and at low elevation coastal regions of Antarctica, though less so than other regional climate models such as RACMO2.3p2[71]. The firn model used by MAR resolves the upper 20 m of firn across 30 layers[33], whereas the CFM resolves between 100 and 200 m of firn across approximately 2000 layers[32]. Therefore, we use the CFM to simulate FAC, runoff and ice lens depth rather than MAR because it is able to resolve snow/firn density changes at a higher vertical resolution and with an increased level of detail in the model physics.

We acknowledge that the current handling of meltwater retention, refreezing and runoff in high-melt areas is currently a limitation of firn models[76]. This limitation is unfortunately inherent to the most state-of-the-art firn models to date. The firn meltwater Retention Model Intercomparison Project (RetMIP) compared densification and meltwater dynamics from simulations of nine different firn models at four weather station sites on the Greenland Ice Sheet[76]. The CFM took an active part in this intercomparison project. RetMIP quantified uncertainty in modelled runoff by using inter-model spread. It was shown that runoff uncertainty was strongly dependent on total melt rates. At the Dye-2 site, the runoff uncertainty was estimated at 13%[76]. Dye-2 shows melt rate conditions (150 m.we yr$^{-1}$) and firn structural features (formation of ice lenses) comparable to conditions on East Antarctic ice shelves. Therefore, we believe that a 13% uncertainty value on CFM-computed runoff is a sensible estimate.

It is important to emphasise that we do not aim to quantify absolute amounts of surface runoff, but rather year-to-year variability in runoff, surface melt, firn air content and minimum ice lens depth to allow us to investigate these as potential controls on the interannual variability in SGL area and volume. For this reason, possible model biases in these variables do not affect our interpretations if these biases are consistent in time.

MAR surface melt has yet to be specifically validated in detail over coastal East Antarctica and model validation is hindered by the scarcity of published observations of surface melt and runoff, particularly around coastal East Antarctica. In the framework of an Antarctic-specific climate model intercomparison, ref. [71] validated MAR's performance relative to observed surface mass balance. On the ice shelves, MAR showed the lowest bias among the six climate models investigated. Also, the ice shelf surface mass balance simulated by MAR was closest to the ensemble mean, thus likely a good representation of current model estimates. We note that the standard deviation of the intercomparison in ice shelf surface mass balance was 77 Gt/yr, corresponding to 19% of the modelled surface mass balance. Although not directly comparable, this figure can be used as an estimate of melt rates model uncertainty on the East Antarctic ice shelves.

## Data availability

The supraglacial lake extents and volumes generated in this study have been deposited in the UK Polar Data Centre (https://doi.org/10.5285/A9F2E4B5-9C2E-4EA5-8C0C-DB5F6585128A). Landsat and Sentinel satellite imagery is freely available via Google Cloud (gs://gcp-public-data-landsat/ and gs://gcp-public-data-sentinel-2/). ERA5 reanalysis data is available from the Copernicus Climate Change Service Climate Date Store (https://cds.climate.copernicus.eu/).

## Code availability

The code for the lake-detection algorithm is available at: https://github.com/mmoussavi/Lake_Detection_Satellite_Imagery/tree/master/Landsat_8. The CFM code is publicly available under the MIT license at https://github.com/ UWGlaciology/CommunityFirnModel (https://zenodo.org/record/3585885#.Yh5ZWBtOmEA). MAR data is freely available at ftp://ftp.climato.be/climato/ckittel/MARv3.11/ and the model code for MARv3.11 is available at https://gitlab.com/Mar-Group/MARv3.7.

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

## Acknowledgements

J.F.A. was funded by the IAPETUS Natural Environment Research Council (NERC) Doctoral Training Partnership (grant number NE/L002590/1). C.R.S. and S.S.R.J. were supported by NERC grant NE/R000824/1. We would like to thank Mahsa Moussavi for helpful discussions and for provision of masked Landsat 8 scenes (see Data Availability). We acknowledge MAR data from Xavier Fettweis (xavier.fettweis@uliege.be) and the Norwegian Polar Institute's Quantarctica package (Matsuoka *et al*., 2021, doi: 10.1016/j.envsoft.2021.105015).

## Author contributions

J.F.A. designed the initial study, undertook the remote sensing data collection, conducted the analysis and led the manuscript writing. V.V. provided the Community Firn Model outputs. All authors (J.F.A., C.S., S.J., R.C., A.A., V.V.) provided input on the research design and interpretation and commented on the manuscript.

## Competing interests

The authors declare no competing interests.
