## [Peer Review File · Nature Communications]

Large interannual variability in supraglacial lakes around East AntarcticaReviewers' Comments:

Reviewer #1:

Remarks to the Author:

This is a novel paper with original findings and will be of great interest to glaciologists studying water on ice masses (valley glaciers, the Greenland Ice Sheet, but especially Antarctic ice shelves). It will be of particular interest to remote sensors, but also to the modelling community, as a significant portion of the paper compares findings from remote sensing observations and analysis with outputs from a Community Firn Model forced by a Regional Climate Model.

It applies an existing threshold-based methodology using Landsat 8 satellite imagery (Moussavi et al, 2020) and calculates area and volume of supraglacial lakes (SGLs) on the main ice shelves around East Antarctica. This is the first time this has been attempted. Furthermore, for the first time it compares the spatial and temporal patterns of SGL characteristics to variables from the CFM and RCM to identify significant correlations, and it develops regression-based equations to predict the former from the latter. The work is carefully carried out and the methodology is, on the whole, sound. The results are interesting (for example, that SGL volume varies interannually by as much as 200% on individual ice shelves and by up to ~72% across the entire ice sheet; and that peak years of SGL volume are asynchronous between ice shelves, including those experiencing similar mean annual surface melt) and the interpretation of them is generally justified. The work is put into the context of the wider relevant literature. The paper is nicely structured and generally very clearly written.

I have just a few moderate suggestions that I think would improve the work if they were addressed, together with some minor queries.

Moderate suggestions

1. The study uses imagery from January only. While I agree this is likely to capture the month that has the greatest amount of ponded water, it is not necessarily the case. Around line 55, could a better justification be given for not including Dec or Feb in the analysis? For the years considered (2014-2020), what month has the highest runoff according to RACMO? And in the Discussion, could the implications of only including Jan in the analysis be given?
2. Could some more consideration be given to the fact that imagery is biased due to availability, esp. due to cloud cover? The Methods and Supp Figure 9 show that availability of imagery in January is quite variable, with 6-10 images per year for Amery and 0-2 images per year for Voyeykov. Could this information be used to assign error bars to the areas and volumes of water calculated, e.g. the volumes and anomalies shown in Fig 2?
3. Why are different ice shelves featured in the different parts of the analysis and therefore on Figs 2, 3 and 4?
4. Could a better explanation of the possible role of some of the climate variables be given? For example, L 125-6. I don't quite follow the point that is being made here - 'less sensitive' than what? Are you saying that you might expect SGL area to correlate inversely with winter snowfall because water could be accommodated within firn pore space? But that you don't find this because this effect is more than compensated for by high temperatures? But where does Srad fit in to this argument? Also L 132-5, I don't quite follow the logic of what is being proposed here. Is it simply that ice shelves where summer temps are closer to zero experience greater correlation between water extent and air temps, but Amery ice shelf where air temps are far below zero, it is net radiation that dominates in explanation of water extent variability?
5. Could there be a better justification for some of the statistical analysis? For example, is the correlation analysis parametric or non-parametric? I assume the former as linear regression is then performed, which requires the samples to be $n > 30$ and to be normally distributed. Is this the case?
6. I think some more thought needs to be given to the role of correlation vs regression. The former should be used to identify associations and r is the statistic to quote, which will be +ve or -ve depending on the direction of the association. Regression is used when you're interested in building

empirical models between an a priori defined dependent variable and independent variable(s). I think you're really interested in the former rather than the latter and yet you quote r^2 rather than r . The direction of the relationship cannot be gleaned from r^2 , which is a measure of how well data fit a regression model.

7. Also, following on from 6, you say on L163 that there are issues of collinearity in your multiple regression relationships. If multiple regression is really justified here, can you deal with this by: a) identifying and removing some of the highly correlated independent variables; b) linearly combining the independent variables; c) performing an analysis designed for highly correlated variables, such as principal components analysis or partial least squares regression?

8. The relationship between SGL volume and firm mentioned on L158-60 is interesting. It implies that you're measuring saturated firm when you refer to SGLs? And yet earlier you say lakes most likely where blue ice forms. Have I understood this correctly?

Minor queries

L20-22. I think this sentence needs changing. The term 'significant relationships' needs explaining. Relationships between what? Also, as written, the sentence implies the paper calculates SGL area and volume for the future, but this is not the case.

L33. Should the reference numbers here include 19 and 20?

L67-8. To avoid repetition, the phrase "values >200%...Some of" could be deleted and the sentence rewritten to say "...ice shelves, with the largest SGL volume anomalies in January..."

L 74-5 "moderate mean annual surface melt rates of ~50-60 mm w.e. yr-1 on all three ice shelves" Where is the evidence for this?

L112. Is this 'close link' expected to be a negative correlation here?

L115. You mention 'mean summer snowfall' here but this wasn't mentioned in the list of variables in the previous paragraph.

L142-3. Does firn saturation prevent meltwater percolation? Or is it that firn saturation is a consequence of reduced percolation? If surface melt rate exceeds infiltration rate, firn will saturate and a water table in firn will rise? Similarly, is it the case that firn saturation will prevent refreezing? Or is it that saturated firn suggests temperature of firn has been brought up to melt temperature, inhibiting immediate refreezing but it could refreeze subsequently?

L145. You say Nov FAC, Jan FAC and DJF FAC are calculated and compared with SGL volume, but why not Dec FAC or Feb FAC?

L154. Because of the confusion between discussing correlation (r) and quoting the coefficient of determination of regression equations (r^2) it's not immediately obvious the nature of the relationships between variables. For example, on L 154 you say SGL volume is strongly correlated with the Nov FAC to DJF surface melt ratio and quote an r^2 of 0.13. But presumably, the correlation is negative? So, when end of winter FAC is low and / or summer melt is high, SGL volume is high? This would seem to be the case from reading on, L172. Quoting r rather than r^2 would be useful.

L196. "...correlations (r^2)..." See above.

Figures

Where are the ice shelf edges and grounding lines taken from? Can this be mentioned in the Fig headings?

Fig 4. What are the grey / blue areas?

References

Check all these.

Refs 19 and 20 repeat 3 and 4.

For ref 25, it should be EPSL not ESPL.

Reviewer #2:

Remarks to the Author:

Summary

This is a nice study of the inter-annual variability of supraglacial lake extent and area on the East Antarctic Ice sheet's ice shelves. Understanding supraglacial lake dynamics on Antarctic ice shelves is important since they have been linked to rapid ice shelf collapse. The East Antarctic ice sheet is the largest ice sheet in the world and therefore has an outsized potential impact on future global sea levels. This study is an impressive analysis of a huge amount of satellite data. I find the manuscript well written but would like some clarifications at a few places. I also think some findings could have been better illustrated in figures and tables, see comments below.

Major comments

1. How much trust can we put in the CFM simulations of mean summer total runoff and ice lens depth, as well as MAR surface melt. Have these models been validated for the study area?
2. I appreciate seeing the scatter plots, but at the same time it is a lot of information and most of it shows relationships that are not statistically significant. It is a lot of info for the reader to sort through. Perhaps the scatterplots could be in the supplementary figures and some more effective summary figure could be in the main manuscript?
3. It would be nice to include a map that shows the extent and number of satellite scenes for the study area so that the reader gets a good grasp of how much data was analyzed where. I assume this varies spatially.

Minor comments

Figure 1: Indicate the spatial extent of the East Antarctic ice sheet and ice shelves as separate from the other parts of Antarctica.

L64: Can you clarify what 72% and 61% refer to. I assume 72% refer to the volume anomaly fluctuations relative to the mean values seen in Figure 2b? But the text does not say these numbers refer to anomalies. Additionally, the volume fluctuations go between $\sim 70\%$ to $\sim 50\%$. In other words, a clarification would be helpful. Aligning the terminology of the figures with the text might be enough, i.e. call it "anomalies" or "fluctuations". The same applies to the text that follow on L67.

L72: It is hard to see if the SGL volumes are asynchronous between the EAIS ice shelves since the figures shows anomalies and the table only list mean values. It would be nice to show the absolute values. Considering adding a line to the Figure 2 plots or add a supplemental figure.

L74-75: It would be nice to see the mean annual melt rates for each ice shelf in relation to the SGL volume/area variability.

L81-83: It is not possible to see that SGL spreads towards the ice shelf calving fronts in Figure 3 since this figure shows reoccurrence frequency, but not when lakes occurred.

L84-87: It is hard to follow why correlations between SGL area and maximum SGL elevation would reveal the elevations where SGL's were formed. More explanation and context are needed.

L87: Maybe cite Figure 4 here

Figure 4: Explain the blue areas (SGL) and grey lines demarking the glacier outlines in the figure (although, perhaps the glacier outlines are not necessary)

L113-114: Clarify if this a correlation between the two variables and SGL or the two variables listed in the sentence. The context suggests SGL, but the next sentence suggests it is the correlation between clouds and net solar radiation. However, the motivation for examining the correlation between clouds and net solar radiation in the context of SGL variability is unclear.

L133: Wouldn't the blue ice effect from katabatic winds also be important here?

L138 Check when FAC is first mentioned and explain it there. For example, I found it on line 124

L165: I am not so surprised since I don't think either MAR or CFM has been validated properly for East Antarctica. Obviously, this is very hard because of the lack of in situ data of runoff, surface melt, and ice lens depth. Perhaps discuss the model uncertainties and lack of validation. I see this mentioned on line 210-212 and 215-216, but could perhaps be mentioned earlier. In other words, did you really expect good agreement before you did the analysis given what we already know about these models and the lack of validation.

Reviewer #1 (Remarks to the Author):

This is a novel paper with original findings and will be of great interest to glaciologists studying water on ice masses (valley glaciers, the Greenland Ice Sheet, but especially Antarctic ice shelves). It will be of particular interest to remote sensors, but also to the modelling community, as a significant portion of the paper compares findings from remote sensing observations and analysis with outputs from a Community Firn Model forced by a Regional Climate Model.

It applies an existing threshold-based methodology using Landsat 8 satellite imagery (Moussavi et al, 2020) and calculates area and volume of supraglacial lakes (SGLs) on the main ice shelves around East Antarctica. This is the first time this has been attempted. Furthermore, for the first time it compares the spatial and temporal patterns of SGL characteristics to variables from the CFM and RCM to identify significant correlations, and it develops regression-based equations to predict the former from the latter. The work is carefully carried out and the methodology is, on the whole, sound. The results are interesting (for example, that SGL volume varies interannually by as much as 200% on individual ice shelves and by up to ~72% across the entire ice sheet; and that peak years of SGL volume are asynchronous between ice shelves, including those experiencing similar mean annual surface melt) and the interpretation of them is generally justified. The work is put into the context of the wider relevant literature. The paper is nicely structured and generally very clearly written.

I have just a few moderate suggestions that I think would improve the work if they were addressed, together with some minor queries.

We would like to thank the Reviewer for their work on our manuscript and their very positive and constructive comments. We have answered their specific comments below (noting that line numbers refer to the revised 'tracked changes' version of the manuscript, with tracked changes visible).

Moderate suggestions

1. The study uses imagery from January only. While I agree this is likely to capture the month that has the greatest amount of ponded water, it is not necessarily the case. Around line 55, could a better justification be given for not including Dec or Feb in the analysis? For the years considered (2014-2020), what month has the highest runoff according to RACMO? And in the Discussion, could the implications of only including Jan in the analysis be given?

We used imagery from the month of January because our main aim is to produce a comprehensive, consistent comparison of maximum total surface meltwater around the EAIS over the seven-year period, rather than focusing on the temporal evolution of meltwater over the entire melt season. Quantifying this interannual variability in the maximum surface meltwater that might be present is an important step in advancing process understanding of which ice shelves may be closer to potential thresholds of meltwater-induced hydrofracturing. Our work also extends previous work by Stokes et al. (2019), which provided the first ice sheet-wide assessment of supraglacial lake areas at the peak of the melt season for a single melt season (January 2017).

We agree that late-onset melting on some ice shelves in some years may result in surface runoff peaking later in the melt season. However, we know that lake areas and volumes from Landsat 8 imagery typically peak between mid-late January during our study period, after which lakes start typically freezing over. Including imagery from February is therefore unlikely to have substantially changed our maximum lake area and volume estimates. Furthermore, RACMO has been shown to simulate peak surface runoff in January between 2005 and 2020 over the Amery Ice Shelf (Tuckett et al., 2021) and several other studies also note that maximum meltwater extent occurs in mid-late January (Langley et al., 2016; Arthur et al., 2020, Dell et al., 2020; Dirscherl et al., 2021).

In addition, including December and February in our analysis would be significantly more computationally expensive (we processed 2175 Landsat 8 images just for the month of January 2014-2020).

We have expanded our justification of using imagery from January on Lines 751 – 766 (Methods).

2. Could some more consideration be given to the fact that imagery is biased due to availability, esp. due to cloud cover? The Methods and Supp Figure 9 show that availability of imagery in January is quite variable, with 6-10 images per year for Amery and 0-2 images per year for Voyeykov. Could this information be used to assign error bars to the areas and volumes of water calculated, e.g. the volumes and anomalies shown in Fig 2?

We agree that cloud cover introduces bias in imagery availability. It is difficult to quantify and assign individual errors to lake area and volume estimates per ice shelf because total uncertainty is comprised of pixel resolution uncertainty, uncertainty in the empirical lake depth retrieval algorithm (which has not yet been explicitly quantified for Antarctica), uncertainty in lake boundaries derived from pixel reflectance thresholding, and varying image availability and spatial coverage around the East Antarctic Ice Sheet margin, but we have modified Supplementary Figure 9 (now Supp. Fig. 11) to highlight the bias introduced by imagery availability, including due to cloud cover. Panels b-q in the Supplementary Figure now show the total number of available scenes partially or fully covering each ice shelf/glacier, as well as the proportion of useable (i.e. cloud-free) scenes from which lakes can be mapped. In addition, Supplementary Figure 13 demonstrates that low SGL occurrence is not simply an artefact of a low number of useable images and that we are still able to map lakes even in regions that experience high cloud cover on some dates in January.

3. Why are different ice shelves featured in the different parts of the analysis and therefore on Figs 2, 3 and 4?

Figure 2 contains all the ice shelves and regions where supraglacial lakes formed in our observation period (2014-2020), with the exception of Voyeykov Ice Shelf and Ingrid Christensen Coast (which we decided not to include in this figure in the interest of space and because comparatively very few supraglacial lakes form on these two ice shelves). For completion, we have now added an additional supplementary figure (Supplementary Figure 4) showing supraglacial lake volume anomalies for Voyeykov Ice Shelf and the Ingrid Christensen Coast. Similarly, Figure 3 intends to highlight examples of supraglacial lake reoccurrence on some of these major ice shelves and outlet glaciers and Figure 4 shows maximum lake

elevations for all regions in which we recorded lakes. We have edited the Figure captions to clarify why different ice shelves are shown in these figures.

4. Could a better explanation of the possible role of some of the climate variables be given? For example, L 125-6. I don't quite follow the point that is being made here - 'less sensitive' than what? Are you saying that you might expect SGL area to correlate inversely with winter snowfall because water could be accommodated within firn pore space? But that you don't find this because this effect is more than compensated for by high temperatures? But where does S_{rad} fit in to this argument? Also L 132-5, I don't quite follow the logic of what is being proposed here. Is it simply that ice shelves where summer temps are closer to zero experience greater correlation between water extent and air temps, but Amery ice shelf where air temps are far below zero, it is net radiation that dominates in explanation of water extent variability?

Snowfall during the preceding winter might be expected to be inversely correlated to SGL volumes, because more snowfall during the preceding winter replenishes firn pore space, so more meltwater can be stored within the firn rather than as SGLs (Ligtenberg et al., 2011). Instead, the weak correlations we show between observed SGL areas, volumes and mean snowfall during the preceding winter (as well as with mean summer snowfall and mean summer net surface solar radiation) indicate that these variables exert a weak control on surface melt availability and hence interannual variability in SGL area and volume. This suggests that other controls (namely mean summer air temperature, summer firn air content and the November FAC-to-DJF surface melt ratio) are more important drivers of interannual variability in SGL area and volume. The weak correlations between observed SGL areas, volumes and mean snowfall during the preceding winter and mean summer snowfall might also be related to drifting-snow transport processes, where high snowfall accumulation gets blown and scoured away by surface winds, preventing firn pore space from being replenished in these regions. We also note that drifting-snow transport processes are not currently included in MAR (see Methods).

Surface net solar radiation represents the difference between downward incoming shortwave radiation and reflected longwave radiation (van den Broeke et al., 2004). Mean summer net surface solar radiation (S_{rad}) might be expected to be positively correlated to SGL volumes because enhanced incoming shortwave radiation (and therefore net S_{rad}) increases melting of the snowpack and the volume of liquid meltwater available to form SGLs (Lenaerts et al., 2017; Turton et al., 2021). The grounding zone of Amery Ice Shelf is subject to low near-surface summer air temperatures, which means surface net solar radiation exerts a stronger control on the interannual variability in SGL areas and volumes on this ice shelf, rather than variability in near-surface summer air temperatures or snowfall. Indeed, increased incoming shortwave radiation has been linked to more extensive SGLs on the Amery Ice Shelf (Dirscherl et al., 2021).

We have expanded our discussion of the role played by snowfall and net surface solar radiation on interannual variability in SGL area and volume on Lines 164 to 187 and 198 to 201.

5. Could there be a better justification for some of the statistical analysis? For example, is the correlation analysis parametric or non-parametric? I assume the former as linear regression is then performed, which requires the samples to be $n > 30$ and to be normally distributed. Is this the case?

We perform parametric correlation analysis because we wanted to test the degree to which different climatic and surface conditions influence interannual variability in total SGL area and volume. Furthermore, we wanted to assess whether these variables simulated by ERA5, the CFM and MAR (namely mean November FAC, mean summer FAC, mean summer total runoff, mean summer shallowest ice lens depth, near-surface 2-m air temperature, skin temperature, net surface solar radiation, mean preceding winter snowfall and mean summer snowfall) can be used as a first-order prediction of SGL area and volume over Antarctic ice shelves, as this is one of the first studies to make such comparisons with this range of variables.

Several previous studies have performed Pearson correlation to assess potential drivers of interannual variability in observed total lake area, including with modelled seasonal snowmelt, surface air temperature, and firn air content (Langley et al., 2016; Arthur et al., 2020; Leeson et al., 2020; Tuckett et al., 2021; Dirscherl et al., 2021). The sample sizes in these correlation analyses are typically limited by the length of the satellite record, as in this study. We note that even if we had extended our observational period back in time to also include the Landsat 7 record (i.e. from 2000 to 2020 rather than from 2014 to 2020), which would require significant computational resources, the number of data points in each correlation would still be < 30 . We have added this explanation to our discussion of our individual linear correlation analysis on Lines 839-841 and 853-867.

6. I think some more thought needs to be given to the role of correlation vs regression. The former should be used to identify associations and r is the statistic to quote, which will be +ve or -ve depending on the direction of the association. Regression is used when you're interested in building empirical models between an a priori defined dependent variable and independent variable(s). I think you're really interested in the former rather than the latter and yet you quote r^2 rather than r . The direction of the relationship cannot be gleaned from r^2 , which is a measure of how well data fit a regression model.

We agree that the direction of correlation should be made explicit and have clarified this throughout the main text by quoting r rather than r^2 . We have also provided Pearson's correlation coefficient (r values) for the whole ice sheet (Supplementary Tables 1 and 2) and for all individual ice shelves (Supplementary Tables 3 and 4).

7. Also, following on from 6, you say on L163 that there are issues of collinearity in your multiple regression relationships. If multiple regression is really justified here, can you deal with this by: a) identifying and removing some of the highly correlated independent variables; b) linearly combining the independent variables; c) performing an analysis designed for highly correlated variables, such as principal components analysis or partial least squares regression?

We performed multiple regression to explore the combination of variables most strongly correlated with total SGL volume and area, but acknowledge there is likely to be a strong multicollinearity caused by the

high number of predictor variables. The main focus of our statistical analysis is to explore the relative importance of each variable on controlling the interannual variability in SGL area and volume, rather than their combined influence. Therefore, we have decided our multiple regression does not substantially add to our conclusions and have now removed discussion of this from the main text and from Supplementary Tables 1 and 2 to avoid confusion.

8. The relationship between SGL volume and firn mentioned on L158-60 is interesting. It implies that you're measuring saturated firn when you refer to SGLs? And yet earlier you say lakes most likely where blue ice forms. Have I understood this correctly?

SGLs can form both over saturated firn and on blue ice. The presence of SGLs is closely linked to the presence of saturated firn, which controls the amount of snowmelt that can percolate downward into the snowpack and be absorbed in the firn layer. Most lakes typically form around or just downstream of blue ice areas, which form once the snow and firn layers are entirely removed by wind erosion, sublimation or melt. Blue ice enhances surface melting through a positive melt-albedo feedback (Kingslake et al., 2017; Stokes et al., 2019) and so locally this surface meltwater accumulates as SGLs. Therefore, SGL volume reflects the firn meltwater storage capacity dictated by the previous winter and the impact of summer melt on lowering firn meltwater retention. We have added a more detailed explanation of this on Lines 177-179 and 236-238.

Minor queries

L20-22. I think this sentence needs changing. The term 'significant relationships' needs explaining. Relationships between what? Also, as written, the sentence implies the paper calculates SGL area and volume for the future, but this is not the case.

This is a useful point, and we have reworded the sentence to: 'However, we find January surface melt and the ratio between November firn air content and summer (December-January-February) melt are important predictors of SGL volume on significant relationships on some potentially vulnerable ice shelves, suggesting large increases in SGL coverage and volume should be expected under future atmospheric warming'.

L33. Should the reference numbers here include 19 and 20?

We agree that the first of these reference (Banwell et al., 2013) is relevant to this sentence as it is a key paper providing observations of hydrofracturing and we have now cited it here. While the second reference (Banwell et al., 2019) provides direct measurements of ice-shelf flexure caused by the filling and draining of lakes by overflow and channel incision, it provides no evidence of hydrofracture-induced cracks or moulins, and so we chose not to cite this paper in this particular sentence.

L67-8. To avoid repetition, the phrase "values >200%...Some of" could be deleted and the sentence rewritten to say "...ice shelves, with the largest SGL volume anomalies in January..."

We have reworded the sentence to: 'Interannual fluctuations in SGL volume are even higher on individual ice shelves, with the largest SGL volume anomalies in January 2020, on the Moscow University (225%), Riiser-Larsen (193%) and Shackleton (111%) ice shelves (Fig. 2c,g,i).'

L 74-5 "moderate mean annual surface melt rates of ~50-60 mm w.e. yr⁻¹ on all three ice shelves" Where is the evidence for this?

The mean annual surface melt rates that we are referring to here are the mean annual 1999-2009 melt rates derived from QuikSCAT radar backscatter from Reference 38 (Trusel et al., 2013) and listed in Table 2. We have now clarified this in this paragraph.

L112. Is this 'close link' expected to be a negative correlation here?

We would expect a negative correlation between supraglacial lake volumes and the preceding winter snowfall, because a greater total amount of accumulated winter snowfall increases the air content of the firn and its capacity for storing surface meltwater. Consequently, this manifests as less surface meltwater accumulation and lower supraglacial lake volumes. We have clarified this by replacing 'close link' with 'negatively correlated'.

L115. You mention 'mean summer snowfall' here but this wasn't mentioned in the list of variables in the previous paragraph.

We included mean January snowfall and mean summer (December-January-February) snowfall in our regression analysis (Supplementary Table 1) and have now added these to the list of variables on Lines 114-117.

L142-3. Does firn saturation prevent meltwater percolation? Or is it that firn saturation is a consequence of reduced percolation? If surface melt rate exceeds infiltration rate, firn will saturate and a water table in firn will rise? Similarly, is it the case that firn saturation will prevent refreezing? Or is it that saturated firn suggests temperature of firn has been brought up to melt temperature, inhibiting immediate refreezing but it could refreeze subsequently?

Continued surface melting over successive melt seasons gradually depletes firn pore space when more pore space is lost by melt and refreezing during densification than is replenished by snowfall (Ligtenberg et al., 2011). At this point, meltwater can no longer percolate into and refreeze within the firn, meaning the firn cannot act as an absorbing buffer (Ligtenberg et al., 2011; Munneke et al., 2014). Meltwater percolation and refreezing into the firn layer can exert a localised warming effect on ice temperatures through the release of latent heat (Hubbard et al., 2016; Stokes et al., 2019). Therefore, firn saturation prevents meltwater percolation and refreezing. When this occurs, the firn is flooded and surface meltwater forms SGLs. We have provided additional explanation of this to main text on Lines 209-216.

L145. You say Nov FAC, Jan FAC and DJF FAC are calculated and compared with SGL volume, but why not Dec FAC or Feb FAC?

The reason we compared mean November FAC with total SGL areas and volumes was to assess the influence of firn air content before the onset of the summer melt season, which begins in November (e.g.

Moussavi et al., 2020). This enabled us to separate the effects of summer melt and firm storage capacity on interannual lake variability. We then compared mean January FAC with total SGL areas and volumes to assess the influence of firm air content at the peak of the melt season, which typically occurs in mid-late January (e.g. Stokes et al., 2019). We also chose to compare mean December-January-February FAC with total SGL areas and volumes to assess the influence of surface melt over the whole summer melt season. The correlation analysis we conducted with these three FAC metrics, alongside the other modelled variables, aims to provide a first exploratory comparison between their year-to-year variability with that of SGL area and volume. We have added a sentence to explain this on Lines 225-227.

L154. Because of the confusion between discussing correlation (r) and quoting the coefficient of determination of regression equations (r^2) it's not immediately obvious the nature of the relationships between variables. For example, on L 154 you say SGL volume is strongly correlated with the Nov FAC to DJF surface melt ratio and quote an r^2 of 0.13. But presumably, the correlation is negative? So, when end of winter FAC is low and / or summer melt is high, SGL volume is high? This would seem to be the case from reading on, L172. Quoting r rather than r^2 would be useful.

We agree that the direction of correlation should be made explicit and have clarified this wherever correlations are discussed in the text. We find SGL volume is negatively correlated with the November FAC- to-DJF surface melt ratio, meaning a lower end-of-winter (November) FAC (higher summer melt) is associated with higher January SGL volumes. We have provided Pearson's correlation coefficient (r values) along with r^2 values for the whole ice sheet (Supplementary Tables 1 and 2) and for all individual ice shelves (Supplementary Tables 3 and 4).

L196. "...correlations (r^2)..." See above.

We have now provided Pearson's correlation coefficient (r values) along with r^2 values throughout the main text and have listed them for the whole ice sheet in Supplementary Tables 1 and 2 and for all individual ice shelves in Supplementary Tables 3 and 4.

Figures

Where are the ice shelf edges and grounding lines taken from? Can this be mentioned in the Fig headings?

The coastline and ice shelf edges datasets shown in Figure 4 are from MEaSUREs (Mouginot et al., 2017). The grounding line dataset is the MEaSUREs grounding line from Rignot et al. (2016). We have added these citations to all relevant Figure captions.

Fig 4. What are the grey / blue areas?

The blue areas are supraglacial lakes and the grey areas are ice shelves and floating glacier tongues. We have now clarified this in the Figure key and caption.

References

Check all these.

Refs 19 and 20 repeat 3 and 4.

Corrected

For ref 25, it should be EPSL not ESPL.

Corrected.

Reviewer #2 (Remarks to the Author):

This is a nice study of the inter-annual variability of supraglacial lake extent and area on the East Antarctic Ice sheet's ice shelves. Understanding supraglacial lake dynamics on Antarctic ice shelves is important since they have been linked to rapid ice shelf collapse. The East Antarctic ice sheet is the largest ice sheet in the world and therefore has an outsized potential impact on future global sea levels. This study is an impressive analysis of a huge amount of satellite data. I find the manuscript well written but would like some clarifications at a few places. I also think some findings could have been better illustrated in figures and tables, see comments below.

We would like to thank the Reviewer for their encouraging comments and work on our manuscript. We have answered their specific comments below (noting that line numbers refer to the revised 'tracked changes' version of the manuscript).

Major comments

1. How much trust can we put in the CFM simulations of mean summer total runoff and ice lens depth, as well as MAR surface melt. Have these models been validated for the study area?

The ArMAP firn densification model (Arthern et al., 2010; Verjans et al., 2019) used in the Community Firn Model has been calibrated and validated to an extensive dataset of 91 firn depth-density profiles, of which >60 are from Antarctica (>30 on the East Antarctic Ice Sheet) and showed good performance in capturing firn air content at these calibration sites (Verjans et al., 2020).

We acknowledge that the current handling of meltwater retention, refreezing and runoff in high-melt areas is currently a limitation of firn models (Vandecrux et al., 2020). This limitation is unfortunately inherent to the most state-of-the-art firn models to date. The firn meltwater Retention Model Intercomparison Project (RetMIP) compared densification and meltwater dynamics from simulations of nine different firn models at four weather station sites on the Greenland Ice Sheet (Vandecrux et al., 2020). The CFM took an active part in this intercomparison project. RetMIP quantified uncertainty in modeled runoff by using inter-model spread. It was shown that runoff uncertainty is strongly dependent on total melt rates. At the Dye-2 site, the runoff uncertainty was estimated at 13%. Dye-2 shows melt rate conditions (150 mwe yr^{-1}) and firn structural features (formation of ice lenses) comparable to conditions on East Antarctic ice shelves. Therefore, we believe that a 13% uncertainty value on CFM-computed runoff is a sensible estimate.

It is important to emphasize that we do not aim to quantify absolute amounts of surface runoff, but rather year-to-year variability in runoff, surface melt, firn air content and minimum ice lens depth to allow us to investigate these as potential controls on the interannual variability in SGL area and volume. For this

reason, possible model biases in these variables do not affect our interpretations if these biases are consistent in time.

MAR surface melt has yet to be specifically validated in detail over coastal East Antarctica and model validation is hindered by the scarcity of published observations of surface melt and runoff, particularly around coastal East Antarctica. In the framework of an Antarctic-specific climate model intercomparison, Mottram et al. (2021) validated MAR's performance relative to observed surface mass balance. On the ice shelves, MAR showed the lowest bias among the six climate models investigated. Also, the ice shelf surface mass balance simulated by MAR was closest to the ensemble mean, thus likely a good representation of current model estimates. We note that the standard deviation of the intercomparison in ice shelf surface mass balance was 77 Gt/yr, corresponding to 19% of the modelled surface mass balance. Although not directly comparable, this figure can be used as an estimate of melt rates model uncertainty on the East Antarctic ice shelves.

We have added these points to the Methods on Lines 880-882 and Lines 912-937.

2. I appreciate seeing the scatter plots, but at the same time it is a lot of information and most of it shows relationships that are not statistically significant. It is a lot of info for the reader to sort through. Perhaps the scatterplots could be in the supplementary figures and some more effective summary figure could be in the main manuscript?

We have simplified Figures 5 and 6 by removing R^2 values and non-significant correlations. Regression statistics for all ice shelves are now included in two new Supplementary Tables (3 and 4).

3. It would be nice to include a map that shows the extent and number of satellite scenes for the study area so that the reader gets a good grasp of how much data was analyzed where. I assume this varies spatially.

We have produced a new supplementary figure (new Supplementary Figure 12) showing the extent of satellite scenes used in this study. We have also modified Supplementary Figure 9 (now Supp. Fig. 11) to show the total number of available scenes partially or fully covering each ice shelf/glacier, as well as the proportion of useable (i.e. cloud-free) scenes from which lakes can be mapped, so that the reader can clearly see the number of satellite scenes used for each area.

Minor comments

Figure 1: Indicate the spatial extent of the East Antarctic ice sheet and ice shelves as separate from the other parts of Antarctica.

We have modified this figure with a bold outline delineating the spatial extent of the East Antarctic Ice Sheet and ice shelves.

L64: Can you clarify what 72% and 61% refer to. I assume 72% refer to the volume anomaly fluctuations relative to the mean values seen in Figure 2b? But the text does not say these numbers refer to anomalies.

Additionally, the volume fluctuations go between $\sim 70\%$ to $\sim 50\%$. In other words, a clarification would be helpful. Aligning the terminology of the figures with the text might be enough, i.e. call it “anomalies” or “fluctuations”. The same applies to the text that follow on L67.

72% refers to the fluctuation in percentage SGL volume anomalies (i.e. percentage of the mean 2014-2020 maximum total lake volume) in Figure 2b, which is also included in the Figure caption. We have now aligned the text with the terminology of the figure by rewording Line 69 to ‘we find that SGL area and volume anomalies fluctuate by up to $\sim 72\%$ and $\sim 61\%$ ’ and by rewording ‘fluctuations’ to ‘anomalies’ on Line 71.

L72: It is hard to see if the SGL volumes are asynchronous between the EAIS ice shelves since the figures shows anomalies and the table only list mean values. It would be nice to show the absolute values. Considering adding a line to the Figure 2 plots or add a supplemental figure.

We have added a new supplementary figure (Supplementary Figure 5) to show the absolute values of maximum total supraglacial lake volumes on the EAIS ice shelves and regions.

L74-75: It would be nice to see the mean annual melt rates for each ice shelf in relation to the SGL volume/area variability.

Mean annual melt rates for each ice shelf are listed in Table 2 alongside mean January 2014-2020 maximum total SGL volume.

L81-83: It is not possible to see that SGL spreads towards the ice shelf calving fronts in Figure 3 since this figure shows reoccurrence frequency, but not when lakes occurred.

This is a good point and we agree that this is not clearly visible from Figure 3. Instead, this is more clearly visible in Supplementary Figure 6 and so we have removed the references to Figure 3 in this sentence to avoid any confusion.

L84-87: It is hard to follow why correlations between SGL area and maximum SGL elevation would reveal the elevations where SGL’s were formed. More explanation and context are needed.

The reason we compare total SGL areas and maximum elevations is to investigate whether SGLs formed at higher elevations further inland during melt seasons with more extensive surface meltwater accumulation. We would perhaps expect SGLs to reach higher maximum elevations during melt seasons with more extensive surface meltwater, but we find this not to be the case. This demonstrates there is not a simple relationship between the two, and may be because SGL extents expand downstream during years of more extensive meltwater (e.g. Supplementary Figure 6). We have added further explanation of this on Lines 94-99.

L87: Maybe cite Figure 4 here

Done.

Figure 4: Explain the blue areas (SGL) and grey lines demarking the glacier outlines in the figure (although, perhaps the glacier outlines are not necessary)

We agree that the basin outlines do not add additional information to this Figure and so we have decided to remove them. In response to a comment from the other reviewer, we have added a key to explain the blue areas, which are supraglacial lakes.

L113-114: Clarify if this a correlation between the two variables and SGL or the two variables listed in the sentence. The context suggests SGL, but the next sentence suggests it is the correlation between clouds and net solar radiation. However, the motivation for examining the correlation between clouds and net solar radiation in the context of SGL variability is unclear.

We have now reworded this sentence to clarify that neither total SGL area or total volume are strongly correlated with mean snowfall during the preceding winter, nor mean summer snowfall, nor mean DJF net surface solar radiation. These sentences now read: 'However, we find low correlations between both total SGL area and volume with the mean snowfall during the preceding winter and with the mean summer snowfall (Supplementary Table 1, Fig. 5e-h). Mean summer net surface solar radiation (S_{rad}) might be expected to be positively correlated to SGL volumes because enhanced incoming shortwave radiation (and therefore net S_{rad}) increases melting of the snowpack and the volume of liquid meltwater available to form SGLs (Lenaerts et al., 2017; Turton et al., 2021). However, we find low negative correlations between both total SGL area and volume with mean DJF S_{rad} ($r \leq -0.31$, $r^2 \leq 0.14$, $p \leq 0.05$, Supplementary Table 1, Fig. 5i-l).'

L133: Wouldn't the blue ice effect from katabatic winds also be important here?

*This is a good point, and the presence of blue ice produced from katabatic wind scouring is certainly important in the grounding zone of the Amery Ice Shelf. We have modified the sentence to read: 'This means intermittent surface melt makes SGL variability sensitive to albedo variations controlled by snowmelt-albedo feedbacks, where snow containing refrozen meltwater **and blue ice exposed by katabatic winds** have a lower surface albedo than fresh snow and therefore absorbs more incoming solar radiation, leading to more surface melt'.*

L138 Check when FAC is first mentioned and explain it there. For example, I found it on line 124

We first define firn air content (FAC) on Line 37 in the Introduction and so we have removed repetition of it here and just kept the acronym where it is mentioned later on in the text.

L165: I am not so surprised since I don't think either MAR or CFM has been validated properly for East Antarctica. Obviously, this is very hard because of the lack of in situ data of runoff, surface melt, and ice lens depth. Perhaps discuss the model uncertainties and lack of validation. I see this mentioned on line 210-212 and 215-216, but could perhaps be mentioned earlier. In other words, did you really expect good agreement before you did the analysis given what we already know about these models and the lack of validation.

We performed regression analysis with each of these modelled variables because we wanted to explore the degree to which different climatic and surface conditions simulated by the CFM and MAR can be used as first-order predictions of interannual variability in SGL area and volume over Antarctic ice shelves.

In our response to the reviewer's first major comment above, we outline how the ArMAP firn densification model (Arthern et al., 2010; Verjans et al., 2019) used in the Community Firn Model has been calibrated and validated to an extensive dataset of 91 firn depth-density profiles, of which >60 are from Antarctica (>30 on the East Antarctic Ice Sheet) and showed good performance in capturing firn air content at these calibration sites (Verjans et al., 2020). We added further detail regarding this on Lines 880-882.

As we discuss above, we acknowledge that the current handling of meltwater retention, refreezing and runoff in high-melt areas is currently a limitation of firn models. This limitation is unfortunately inherent to the most state-of-the-art firn models to date. It is important to emphasize that we do not aim to quantify absolute amounts of surface runoff, but rather year-to-year variability in runoff, surface melt, firn air content and minimum ice lens depth to allow us to investigate these as potential controls on the interannual variability in SGL area and volume. For this reason, possible model biases in these variables do not affect our interpretations if these biases are consistent in time. We have now added this to the Methods section on Lines 912-937, together with model uncertainty values derived from the firn meltwater Retention Model Intercomparison Project (RetMIP) and MAR climate model intercomparison.

Reviewers' Comments:

Reviewer #1:

Remarks to the Author:

I am very pleased to see the authors have responded fully to my initial review comments and I am very satisfied with their responses and changes they've made to the manuscript accordingly. I think the paper is stronger and more readable as a result, particularly having removed the multiple regression work.

I have just minor points and a suggestion to reference a new relevant paper.
Check references. 21 is quoted before 20, 24 before 23. There may be others not in chronological order.

L 116-18. You mention other winter snow variables here but these not shown in Table 2 in Methods or Fig 5 as implied.

L198 singular 'absorb'

L286. You say "low correlations with modelled FAC-to-melt ratio ($r = -0.42$, $r^2 = 0.18$, $p = 0.22$)" but above (L226-228) you said:

"Across the whole ice sheet, total SGL volume is most strongly correlated with mean summer FAC ($r = -0.37$, $p < 0.001$) and with the November FAC-to-DJF surface melt ratio ($r = -0.05$, $p < 0.01$) (Supplementary Table 2)."

-0.42 seems quite high (higher than -0.05) but the point is one is insignificant and the other is significant?

L660 & 671 & 686 I think it is more grammatically correct to say "occur with less frequency"

Fig 4. Can you write in Fig caption more precisely what is meant by the lakes shown in blue? Is this a 7-year composite of lake extent?

Figs 5 & 6. In the caption it's not so much the signif. relationships are displayed in bold, just that they are displayed, i.e. non signif. relationships are not displayed.

Supp Material

L202 need to refer to green and blue not bold for signif. relationships.

Is a correlation of -0.05 really significant?

Since the first review of the paper, there's been another paper published which is related to this work and which the authors may wish to reference and compare some results with.

Dell, R., Banwell, A., Willis, I., Arnold, N., Halberstadt, A., Chudley, T., & Pritchard, H. (2021). Supervised classification of slush and ponded water on Antarctic ice shelves using Landsat 8 imagery. *Journal of Glaciology*, 1-14. doi:10.1017/jog.2021.114

For example, the new Dell et al paper applies a different method (clustering & supervised classification) to the one used here (thresholding) and maps slush as well as water. Comparing Dell et al Fig 5 with Fig 2 of this paper and looking at Roi Baudouin Ice Shelf, there are some interesting comparisons. The anomalously high year of 2017 and low year of 2019 seem to be captured by both methods for example. There are some subtle differences in 2016 and 2018 though.

Response to Reviewer

I am very pleased to see the authors have responded fully to my initial review comments and I am very satisfied with their responses and changes they've made to the manuscript accordingly. I think the paper is stronger and more readable as a result, particularly having removed the multiple regression work. I have just minor points and a suggestion to reference a new relevant paper.

We would like to thank the Reviewer for their work on our revised manuscript and their second round of comments, which are much appreciated. We have answered their specific comments below (noting that line numbers refer to the revised 'tracked changes' version of the manuscript).

Check references. 21 is quoted before 20, 24 before 23. There may be others not in chronological order.

We have amended the order of these references and verified that all other references are in chronological order.

L 116-18. You mention other winter snow variables here, but these not shown in Table 2 in Methods or Fig 5 as implied.

We have now quoted Table 2 earlier in the sentence (Line 107) and added references to Supplementary Tables 1, 4 and Figure 9 (Lines 108-9), where these snow variables are presented.

L198 singular 'absorb'

This is now amended.

L286. You say "low correlations with modelled FAC-to-melt ratio ($r = -0.42$, $r^2 = 0.18$, $p = 0.22$)" but above (L226-228) you said:

"Across the whole ice sheet, total SGL volume is most strongly correlated with mean summer FAC ($r = -0.37$, $p < 0.001$) and with the November FAC-to-DJF surface melt ratio ($r = -0.05$, $p < 0.01$) (Supplementary Table 2)."

-0.42 seems quite high (higher than -0.05) but the point is one is insignificant and the other is significant?

We went back and verified all of our regression statistics and found an error in our reported values for the regression between total SGL volume and the November FAC-to-DJF surface melt ratio (Supplementary Table 2). Whilst we had correctly reported a very weak negative correlation ($r = -0.05$), the r^2 should be 0.003 (not 0.13) and the p -value should be 0.60 (not 0.004), i.e. not significant. Similarly, for total SGL area, we had correctly reported a weak negative direction of correlation ($r = -0.07$), but the r^2 should be 0.004 (not 0.19) and the p -value should be 0.54 (not 0.0003), i.e. not significant. We have amended this in Supplementary Table 2 and lines 188-194 of the main text.

This does not change the main findings of our paper, because modelled January melt and the ratio of November firn air content to summer melt are still important predictors of SGL volume on the two ice shelves where SGL volumes are significantly positively correlated with modelled surface runoff (Shackleton Ice Shelf and West Ice Shelf).

L660 & 671 & 686 I think it is more grammatically correct to say "occur with less frequency"

This has now been amended in the figure captions.

Fig 4. Can you write in Fig caption more precisely what is meant by the lakes shown in blue? Is this a 7-

year composite of lake extent?

In Figure 4, the lakes in blue show lake extents in January 2017, which was the most extensive lakes year and which we choose simply to visualise their distribution. This has now been added to the Figure caption.

Figs 5 & 6. In the caption it's not so much the signif. relationships are displayed in bold, just that they are displayed, i.e. non signif. relationships are not displayed.

Both Figure captions have now been amended to delete 'in bold' to make this clear.

Supp Material

L202 need to refer to green and blue not bold for signif. relationships.

The caption of Supplementary Tables 3 and 4 has now been amended to refer to blue for significant relationships, and these have been shaded accordingly in both tables.

Is a correlation of -0.05 really significant?

Our updated regression statistics show that total SGL volume is not correlated with the November FAC-to-DJF surface melt ratio at the ice-sheet scale ($r = -0.05$, $r^2 = 0.003$, $p\text{-value} = 0.60$). Likewise, total SGL volume is not correlated with the November FAC-to-DJF surface melt ratio at the ice-sheet scale (i.e. when included total SGL volumes for data aggregated across the whole ice sheet for all ice shelves together). We have amended this in Supplementary Table 2 and lines 188-194 of the main text. As above, this does not change the main findings of our paper, because modelled January melt and the ratio of November firn air content to summer melt are still important predictors of SGL volume on the two ice shelves where SGL volumes are significantly positively correlated with modelled surface runoff (Shackleton Ice Shelf and West Ice Shelf).

Since the first review of the paper, there's been another paper published which is related to this work and which the authors may wish to reference and compare some results with (Dell, R., Banwell, A., Willis, I., Arnold, N., Halberstadt, A., Chudley, T., & Pritchard, H. (2021). Supervised classification of slush and ponded water on Antarctic ice shelves using Landsat 8 imagery. *Journal of Glaciology*, 1-14).

For example, the new Dell et al paper applies a different method (clustering & supervised classification) to the one used here (thresholding) and maps slush as well as water. Comparing Dell et al Fig 5 with Fig 2 of this paper and looking at Roi Baudouin Ice Shelf, there are some interesting comparisons. The anomalously high year of 2017 and low year of 2019 seem to be captured by both methods for example. There are some subtle differences in 2016 and 2018 though.

We thank the reviewer for suggesting this paper, which we agree is relevant to our manuscript and provides interesting comparisons with our mapped lake areas and volumes. We have added in a sentence on Line 78, which reads 'Years of peak SGL volume in 2017 and minimum SGL volume in 2019 on the Roi Baudouin Ice Shelf (Fig. 2) are consistent with years of maximum and minimum meltwater and slush extents derived from supervised classification of Landsat 8 imagery³⁸.'